# Glacier-permafrost relations in a high-mountain environment: Five decades of kinematic monitoring at the Gruben site, Swiss Alps

Isabelle Gärtner-Roer[1], Nina Brunner[1], Reynald Delaloye[2], Wilfried Haeberli[1], Andreas Kääb[3], and Patrick Thee[4]

[1]Department of Geography, University of Zurich, Zurich, 8057, Switzerland
[2]Department of Geosciences, University of Fribourg, Fribourg, 1700, Switzerland
[3]Department of Geosciences, University of Oslo, Oslo, 0371, Norway
[4] Swiss Federal Institute for Forest, Snow and Landscape Research (WSL), Zurich, 8903, Switzerland

*Correspondence to*: Isabelle Gärtner-Roer (isabelle.roer@geo.uzh.ch)

**Abstract.** Digitized aerial images were used to monitor the evolution of perennially frozen debris and polythermal glacier ice at the intensely investigated Gruben site in the Swiss Alps over a period of about 50 years. The photogrammetric analysis allowed for a compilation of detailed spatio-temporal information on flow velocities and thickness changes. In addition, high-resolution GNSS (Global Navigation Satellite System) and ground-surface temperature measurements were included in the analysis to provide insight into short-term changes. Over time, extremely contrasting developments and landform responses

are documented. Viscous flow within the warming and already near-temperate rock glacier permafrost continued at a constant average but seasonally variable speed of typically decimetres per year, with average surface lowering limited to centimetres to few decimetres per year. This constant flow causes the continued advance of the characteristic convex, lava stream-like rock glacier with its over-steepened fronts. Thawing rates of ice-rich perennially frozen ground to strong climate forcing are very low (centimetres per year) and the dynamic response strongly delayed (time scale decades to centuries). The adjacent cold

debris-covered glacier tongue remained an essentially concave landform with diffuse margins, predominantly chaotic surface structure, intermediate thickness losses (decimetres per year) and clear signs of down-wasting and decreasing flow velocity. The former contact zone between the cold glacier margin and the upper part of the rock glacier with disappearing remains of buried glacier ice embedded on top of frozen debris exhibits complex phenomena of thermokarst in massive ice and backflow towards the topographic depression produced by the retreating glacier tongue. As is typical for glaciers in the Alps, the largely

debris-free glacier part shows a rapid response (time scale years) to strong climatic forcing with spectacular retreat (>10 meters per year) and mass loss (up to >1 meter water equivalent specific mass loss per year). The system of periglacial lakes shows a correspondingly dynamic evolution and had to be controlled by engineering work for hazard protection.

## 1 Introduction

High alpine environments are characterised by perennial surface and subsurface ice, typically found in glaciers and permafrost.
Glacier and permafrost related processes can interact in a number of ways (Haeberli, 2005). Both, glaciers and permafrost are

important for landscape evolution, the hydrological cycle, the mountain sediment budget, the stability of mountain slopes and associated natural hazards. Due to their characteristic thermal conditions, close to melting or thawing temperature, the occurrence and preservation of glaciers and permafrost is strongly affected by atmospheric temperature rise (IPCC SROCC, 2019) which appears to be stronger in cold mountain areas than on a global average (UNEP, 2007; MRI, 2015).

Mountain glaciers and their fluctuations are recognised as key indicators of climate change; their mass balance primarily reflects a direct response to changing atmospheric conditions, while their volume and length changes (advance/retreat) represent an indirect, delayed and filtered signal (Zemp et al., 2015). Debris cover is present on 44% and prominent (covering > 1 km$^2$) on 15% of the glaciers worldwide (Herreid and Pellicciotti, 2020). Glaciers with a thick debris cover are limited in terms of their use as climatic indicators, since the debris cover influences the energy balance, delays the dynamic response,

and influences the ablation rate as well as the discharge of melt water (Nakawo et al., 2000; Reid and Brock, 2010; Ragettli et al., 2015; Ayala et al., 2016). When melting rapidly, glaciers waste down or back where they have a clean surface, while changes in the debris-covered part typically are significantly smaller and the processes more complex (Benn et al., 2012; Mölg et al., 2020). Corresponding process differences can, for instance, enhance the potential of lake formation in the contact zone of clean and debris-covered ice and may cause glacier lake outburst floods (Kääb et al., 2005; Benn et al., 2012).

Together with long-term measurements of borehole and near-surface temperatures, the monitoring of changes in ice-rich perennially frozen debris and related viscous flow – here called "permafrost creep" concerning the process, and called "rock glacier" concerning the resulting landform – is a key element within long-term observation programmes for mountain permafrost such as PERMOS (2019) in the Swiss Alps. In accordance with the IPA Action Group "Rock glacier inventories and kinematics" (RGIK, 2020), rock glaciers as landforms are defined here by their characteristic morphology exhibiting long-

term cohesive creep with reduced internal friction, their convex shape with over-steepened fronts resulting from continued advance, their composition of talus or debris with variable but generally high (excess) ice contents, as well as by their complex flow behaviour as affected by negative subsurface temperatures and highly anisotropic material properties (Wahrhaftig and Cox, 1959; Haeberli, 1985; Haeberli et al., 1998; Florentine et al., 2014; Merz et al., 2015, 2016). Typically, rock glaciers creep downslope with velocities of several decimetres up to some meters per year (e.g., Roer, 2007, Bodin et al. 2009, Fleischer

et al., 2021; Kääb et al., 2021). This creep behaviour results from internal deformation of the frozen debris and shearing within narrow horizons at typical depths of around 20 m (Arenson et al., 2002; Cicoira et al., 2021). Reliable information on the kinematics of rock glaciers provides insights into the evolution of ice-rich permafrost on mountain slopes and facilitates the analysis of its dynamics (Roer et al., 2005b; Kääb et al., 2007). Therefore, terrestrial as well as remote sensing techniques are applied to quantify the kinematics of permafrost creep, i.e. horizontal surface velocities and vertical surface changes,

respectively (Lambiel and Delaloye, 2004; Kääb, 2005; Haeberli et al., 2006; Roer, 2007; Bodin et al., 2009; Strozzi et al., 2020; Cusicanqui et al. 2021; Fleischer et al., 2021: Kääb et al., 2021; Kaufmann et al., 2021).

In regions with moderately continental climatic conditions, contacts between polythermal glaciers and permafrost are widespread. Investigating the full spatial-temporal complexity of relations and interactions between surface and subsurface ice goes far beyond speculative landform interpretation from visual inspection alone. It requires the application of sophisticated

quantitative methodologies such as drilling, geophysical soundings, temperature recording and geodetic/photogrammetric observation in order to define material characteristics, physical conditions and related processes (e.g., Reynard et al., 2003; Haeberli, 2005; Kneisel and Kääb, 2007; Bosson et al., 2014; Monnier et al., 2014; Janke et al., 2015; Bolch et al., 2018; Kunz and Kneisel, 2020; Falatkova et al., 2020; Kunz et al., 2021; Robson et al., 2021; Vivero et al., 2021). This can become important also for applied purposes in the context of formation and evolution of potentially hazardous periglacial lakes. Due

to repeated historical outburst floods from such lakes in an environment with interacting glaciers and permafrost, the Gruben cirque (Fig. 1) in the Saas Valley, Swiss Alps, has been intensely investigated over many years and using numerous comprehensive field measurements, many of them in connection with protective construction work carried out on behalf of the responsible political authorities and the Swiss National Science Foundation (National Research Programme 31; Haeberli and Röthlisberger, 1976; Röthlisberger, 1979; Haeberli et al., 2001; Kääb and Haeberli, 2001).

To monitor the evolution of the formerly hazardous periglacial lakes, special large-scale aerial photographs were taken annually by the Federal Office of Topography swisstopo since 1970. A first detailed aero-photogrammetric analysis of the flow field caused by permafrost creep at Gruben rock glacier and its former contact zone with Gruben glacier was published by Haeberli et al. (1979). A detailed analysis of the vertical and horizontal changes at the rock glacier surface between 1970 and 1995 was given by Kääb et al. (1997). Results from numerous geophysical soundings (seismic refraction, ice-penetrating

radar, geo-electrical resistivity, gravimetry), permafrost mapping using the BTS method (Bottom Temperature of winter Snow cover; Haeberli, 1973; Lewkovicz and Ednie, 2004), core drilling, and lichenometric studies all document conditions in the periglacial areas and nearby glacier forefields (Barsch et al., 1979; Haeberli et al., 1979; Haeberli, 1979, 1985; King et al., 1987; Haeberli et al., 2001; cf. supplementary material S1 and S2). The monitoring of kinematics and ground-surface temperatures (GST) on the Gruben rock glacier is systematically documented within the framework of the Swiss Permafrost

Monitoring Network (PERMOS - http://www.permos.ch/) since 2012 and 2015, respectively. On the directly adjacent Gruben glacier, radio-echo soundings, numerous hot-water drillings with borehole observations (englacial temperature, glacier bed-resistivity, subglacial water pressure) and aerophotogrammetric observations on thickness changes provided rich information reported by Haeberli (1976), Haeberli and Fisch (1984), Haeberli et al. (2001), and Kääb (2001). Comprehensive geophysical and dynamic investigations also concerned the steeper part of Gruben glacier above its flat tongue (Kulessa, 2009).

The purpose of the study presented here is (i) to extend the kinematic monitoring series on the tongue of Gruben glacier and the adjacent Gruben rock glacier by another 21 years to describe the observed horizontal and vertical changes for a 46-year period (1970 - 2016; additional in-situ monitoring on the rock glacier provides information until 2020), (ii) to analyse and compare the creep characteristics of the perennially frozen rock glacier ("periglacial part") with its striking oversteepened fronts and "organized" surface structure (longitudinal ridges), and the former contact zone with the polythermal LIA glacier

("glacier-affected part") with its more chaotic surface morphology, and (iii) to analyse and compare the morphological structures and flow of the rock glacier and the debris-covered polythermal glacier adjacent to it in order to differentiate the characteristics of the two landforms and their evolution in time. In sum, our study aims to analyse a 50-year time series of

interconnected periglacial, paraglacial and glacial processes and their morphological expressions in a cold, high mountain environment.

## 2 The Gruben site

The polythermal Gruben glacier and the adjacent active rock glacier are situated below the western face of Fletschhorn (3993 m a.s.l.) in the canton of Valais, southern Swiss Alps (46° 10'N / 7° 58' E; Fig. 1). The southern Valais, in between the Great St. Bernhard, the Rhone valley and the Simplon area is formed by the St. Bernhard and Monte Rosa penninic nappes. The Gruben site is characterised by the Siviez - Mischabel sub-nappe, where polymetamorphic rocks dominate (Labhart, 1998). The climate of the study area is predominantly influenced by air masses from southwest and has an inner-Alpine, moderately continental character. According to meteorological measurements at high elevations in the Swiss Alps (MeteoSwiss) and applying an environmental lapse rate of -0.60° C/100 m, mean annual air temperature at the site (2700 – 3000 m a.s.l.) can be estimated at some -2 to -4° C for the time period 1980 - 2009. Around the end of the Little Ice Age, temperatures have been colder by about 1° C (Ilyashuk et al., 2019), whereas the last decade (2011 - 2020) warmed by about 0.8° C (in comparison to 1980 – 2009). Regional precipitation has not been measured on site, but is probably around or even slightly below 1000 mm per year as the Gruben area is particularly shielded by high-mountain ridges in all directions within a few tens of kilometres.

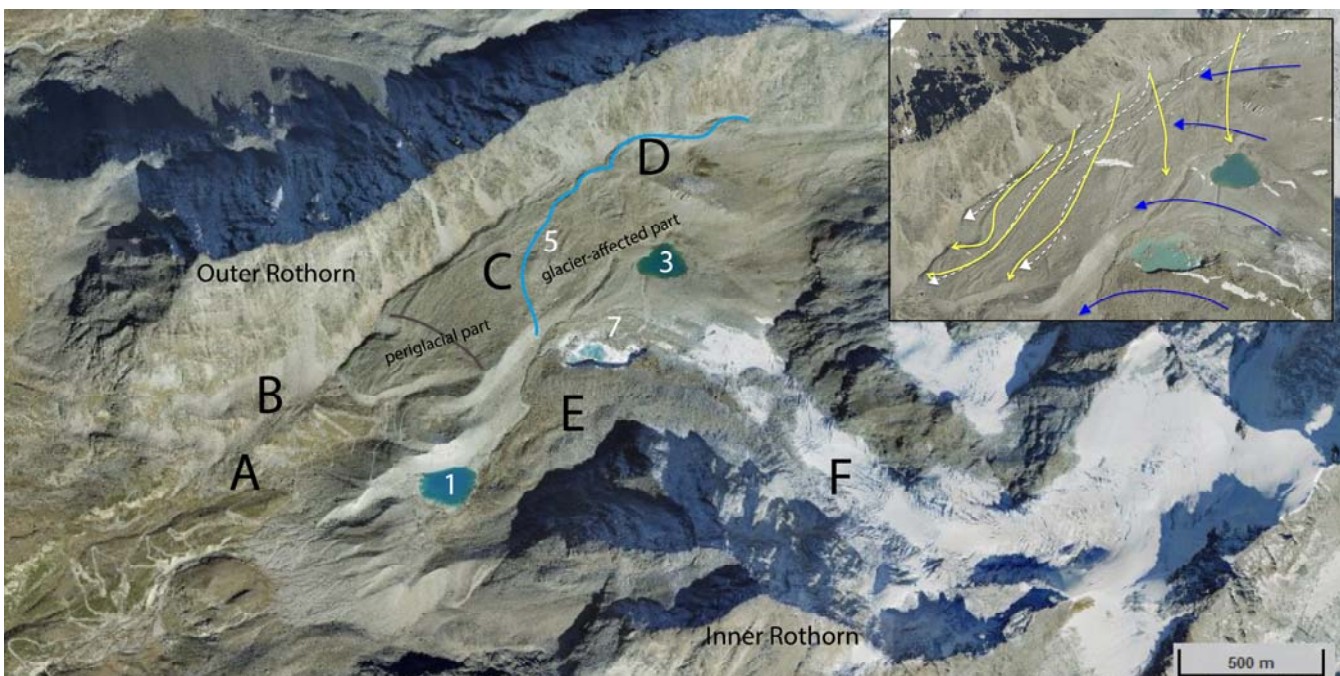

**Figure 1: Geomorphological description of the Gruben site as shown on an orthophoto from 2017 (© SWISSIMAGE, geodata@swisstopo) The blue line indicates the approximate outline of Gruben glacier in the contact zone with the Gruben rock glacier at the time of the Little Ice Age (LIA). The grey line on the rock glacier indicates the position of a seismically determined**

subsurface bedrock riegel. Landforms: A = inactive but still frozen rock glacier; B = actively creeping, frozen protalus rampart; C = Gruben rock glacier; D = deformed frozen talus/moraine; E = debris-covered tongue of Gruben glacier; F = Gruben glacier; 1, 3, 7 = existing lakes; 5 = former thermokarst lake (lake numbering follows historical scheme in order to keep lake identities constant over time; cf. Figure 4). The inset in the upper right (see full Figure 1 for scale) shows the flow trajectories in frozen debris as determined by aero-photogrammetry with the yellow lines (cf. Figure 14 in Haeberli et al., 1979; cf. supplementary material S3) and flow trajectories in frozen debris (white dashed lines) which follow the surface structures produced by long-term cumulative deformation of the frozen talus. The blue arrows indicate the estimated flow direction of the LIA glacier as derived from the earliest reliable topographic map (1889; cf. supplementary material S4, S5 and S6).

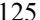

Figure 2: Permafrost distribution at the Gruben site as derived from the Alpine Permafrost Index Map (Böckli et al., 2012), overlay on GoogleEarth (© Google Earth 2009; cf. supplementary material S7 and S8 for comparison with other spatial permafrost simulations at the site). Thin black line indicates the outline of the rock glacier. White dashed lines indicate transition to permafrost-free terrain as documented by BTS measurements and geophysical soundings (cf. supplementary material S1).

Under such cold-dry conditions, mountain permafrost is widespread (Fig. 2; cf. other model results for the site from FOEN, 2005 and Kenner et al., 2019 (S1 and S2)), which are in general agreement with results from field measurements and also with large-scale simulations at 1 km resolution by Obu et al., 2019 (Northern Hemisphere) or Gruber, 2012 (worldwide)). Especially
cold local microclimates thereby exist for two situations: (a) in steep rock faces with thin winter snow and exposed away from the sun, and (b) below surfaces with large blocks where the advection of latent heat by burial of snow in widely-open pore spaces together with efficient convective ventilation in wintertime can locally reduce mean ground temperatures by several centigrades (Hanson and Hoelzle, 2002). Warm conditions relate to more fine-grained surface materials in topographic depressions with thick accumulation of winter snow (e.g. Schneider et al., 2013). Conditions at the Gruben site closely
correspond to this differentiation: the forefields of the polythermal but largely warm-based glaciers, with their finer materials and flat to partly even concave topography are permafrost-free as indicated with white dashed lines in Fig. 2, while the shady northwest-oriented rock faces below the Inner Rothorn and the blocky "foot-of-talus" situations below the Outer Rothorn are perennially frozen. The constituting lithic materials of the rock glacier are derived from the long-term (i.e. Holocene) erosion of the headwall reaching from Outer Rothorn to Senggchuppa and through various sediment transfer processes varying over
time. Borehole temperatures at 3 m depth close to but below the permafrost table in the upper part of Gruben rock glacier (cf. supplementary material S2 and S3) were about -1° C from 1977 - 1982 and BTS values were about -5° C (Haeberli, 1985). Recent GST values at the rock glacier surface (Fig. 3) document that the rock glacier permafrost was still thermally active in 2015 - 2020, with an active layer regularly freezing through during wintertime as documented by near-consistently cold BTS values.
The surface thermal behaviour at the Gruben rock glacier is well in accordance with the observations on other permafrost sites in the region (Valais – Cervinia – Furka - Gotthard), which show a consistent warming trend by about +0.36° C/decade over the last 20 years (Fig. 3; PERMOS, 2020). With mean annual near-surface temperatures close to and even above 0° C, the thickness of the frozen materials reaching up to about 100 m must be inherited from colder phases of the Holocene and the Little Ice Age (cf. Haeberli, 1985; Haeberli et al., 2001) and is not in equilibrium with today's climate. In view of pronounced
thermal offsets within the active layer (ventilation, balch effect) and at the permafrost table (latent heat), mean annual permafrost temperatures at the depth of zero annual amplitude (about 15 m) can be expected to be close to thawing conditions but still slightly negative even today.

Viscous creep of perennially frozen talus produces a large number of active rock glaciers in the region (Frauenfelder, 1998; Barboux et al., 2015). Smaller permafrost landforms, such as an actively moving protalus rampart and an inactive but still
frozen rock glacier, exist at Gruben down to elevations around 2600 m a.s.l. Cohesive deformation of frozen talus/right-lateral moraine is also indicated in the uppermost part of the cirque (Fig. 1; Haeberli, 1979). From the electrical resistivities (high kΩm-range), the strong attenuation of electromagnetic waves, the high but variable P-wave velocities, frontal advance rates of the rock glacier (Kääb and Reichmuth, 2005), and shallow core drilling (7 m) on top of it, overall volumetric ice contents of the ice-supersaturated frozen talus material are estimated to be within about 60 to 85% by volume (cf. similar values reported
from geophysical soundings and core drillings at comparable sites by Florentine et al., 2014; Krainer et al., 2014; Merz et al.,

2015, 2016). In the periglacial part of Gruben rock glacier there is no measured evidence of major buried ice bodies (Haeberli, 2021). Extrapolating back in time, the photogrammetrically determined advance rate of 0.15 times the surface velocity at the front, the development of Gruben rock glacier can be estimated to have taken place during major parts of the Holocene, i.e. over millennia (Kääb et al., 1997). This is in accordance with absolute age determinations by exposure, luminescence or

radiocarbon datings at other rock glaciers in the Alps (Haeberli et al., 2003; Fuchs et al., 2013; Krainer et al., 2014; Amschwand et al., 2021; cf. also Nesje et al., 2021 concerning a low-elevation case in Norway). The Gruben rock glacier front now advances over vegetation-covered, permafrost-free terrain.

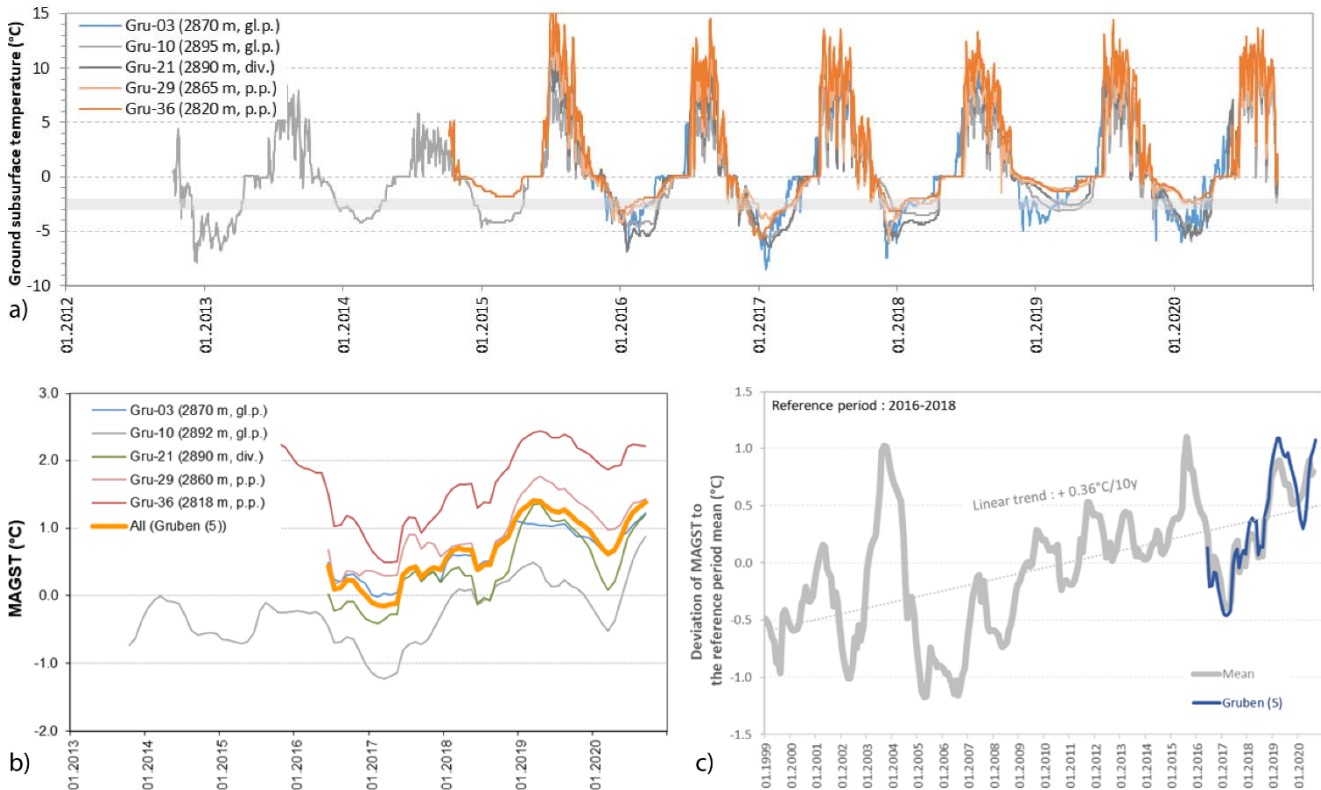

**Figure 3: Daily ground surface temperature on the Gruben rock glacier (gl.p.: glacier-affected part; p.p.: periglacial part; div.:**
**transition zone) since 2013, with indication of the upper limit of BTS for permafrost conditions (horizontal light-grey bar) (a), mean annual ground surface temperature (MAGST) (b), and deviation to the reference period 2016 - 2018 in comparison with the overall signal measured on rock glaciers and debris landforms (n= 2 to >20) in the surrounding region (c) (data: PERMOS and University of Fribourg).**

*Gruben glacier* flows down from the Fletschhorn in the shape of a mirrored "S" and has its active tongue at about 2880 m a.s.l., while its *debris-covered* part on the orographic left side reaches down to 2780 m a.s.l. (see Fig. 1). The glacier was polythermal when investigated in the 1970s with borehole temperatures. Firn temperatures were close to -10° C in its highest parts, temperate firn existed in the steep lower accumulation area, and the tongue was partially cold with 10 m-temperatures

of -1 to -2° C (Haeberli, 1976). The rapidly vanishing flat part of the ablation area was frozen to its bed at the margins but otherwise warm-based; it rested on relatively fine sandy sediments exceeding in places a thickness of 100 m (Haeberli and Fisch, 1984). Artesian water has been observed in hot-water drillings at the transition zone between warm- and cold-based ice (Haeberli et al., 1992, 2001). The artificial tunnel through the cold glacier margin at lake 3 established after the outburst events in 1968 and 1970 for lake-level lowering had been carved into the frozen part of this subglacial bed material. The orographic left part of the tongue can be assumed to be somewhat colder than the rock glacier, because it receives more shadow from the Inner Rothorn. It is heavily covered with debris from intense rock-fall activity in the rock walls to the south of it, which are affected by strong glacial de-buttressing and probably also by permafrost degradation.

During cold periods of the Holocene, especially during historical advances, the upper part of the rock glacier was partly in contact with the orographic right margin of the polythermal Gruben glacier, which deposited some debris-covered ice (up to about 20 meters thick; Kääb and Haeberli, 2001) on top of the permafrost. This complex part of the rock glacier is called the *glacier-affected* part, whereas the lower part is described as the *periglacial* part (Kääb et al., 1997; see Fig. 1 and 4), both being separated by a diffuse transition zone without any clear geomorphological limit. The position of this somewhat diffuse contact zone is defined by (a) the clear margin of the debris-free glacier as indicated in the first reliable topographic map (1889; cf. supplementary material S4, S5 and S6), (b) the limits of exposed massive ice as documented on the annually flown airphotos, and (c) the direction of the flow trajectories which lead from the talus at Outer Rothorn to the rock glacier front; see supplementary material S3. The exposed buried ice on top of permafrost in the former contact zone seems to have vanished during the past years.

Between 1970 and the mid 1990s, changes in geometry and movement have been pronounced for the clean part of Gruben glacier, minimal for the periglacial part of Gruben rock glacier and intermediate for the glacier-affected part of the rock glacier (Haeberli et al., 2001; Kääb, 2001), indicating different dynamics of glacial, periglacial and paraglacial processes.

Since the early 20[th] century, several periglacial lakes had developed and in cases disappeared again in the Gruben cirque at elevations between 2770 and 2900 m a.s.l. (lakes 1, 3, 5 and 6 in Fig. 1 and 4). Lake outbursts and associated floods and debris flows through the Fällbach Creek had repeatedly threatened and damaged the village of Saas Balen (1500 m a.s.l.) in the Saas Valley. Lake 1 is a moraine-dammed lake in front of the debris-covered glacier tongue of Gruben glacier (proglacial lake). In the years 1968 and 1970, outburst floods in combination with the sudden emptying of Lake 3 into Lake 1 formed a deep breach in the large moraine threshold and developed into devastating debris flows. With the construction of a reinforced and controllable outlet structure, Lake 1 was later turned into a retention basin with a capacity of about 100,000 m$^3$ for floods from the upper parts of the cirque (Haeberli et al., 2001). Lake 3 was long situated directly at the cold margin of the glacier tongue at 2860 m a.s.l. Following the outbursts of 1968 and 1970, an artificial outflow tunnel through ice and subglacial permafrost of the glacier margin was constructed for lake-level regulation. This tunnel was replaced in 1996 by an open channel along the retreating glacier margin. In the following year, continued and even enhanced thinning and melting back of the glacier margin eliminated direct ice contact of the remaining water in the lake, the level of which was further lowered in 2003 through a

shallow artificial cut in morainic material of the former glacier bed. Lake 5 – a classical thermokarst lake – started developing in the 1960s in buried massive ice on top of the rock glacier in its glacier-affected part. It continually migrated with, and grew on top of, the slowly creeping permafrost underneath (Kääb and Haeberli, 2001). In autumn 1994 the lake had a surface of about 10,000 m$^2$ and was filled with up to 50,000 m$^3$ of water. Due to its increasing hazard potential, the lake was artificially emptied in 1995; partially by pumping and by draining trough an excavated trench. Lakes 2, 4 and 6 were temporary and of less importance with respect to geomorphological and hazard considerations. In 2016, lake 5 was still empty and also lake 3 was of less concern due to the retreating Gruben glacier. A new proglacial lake (lake 7) formed in the connection zone between the debris-free ice and the debris-covered part of Gruben glacier, building a 20 m high ice cliff at its southern bank (Fig. 4).

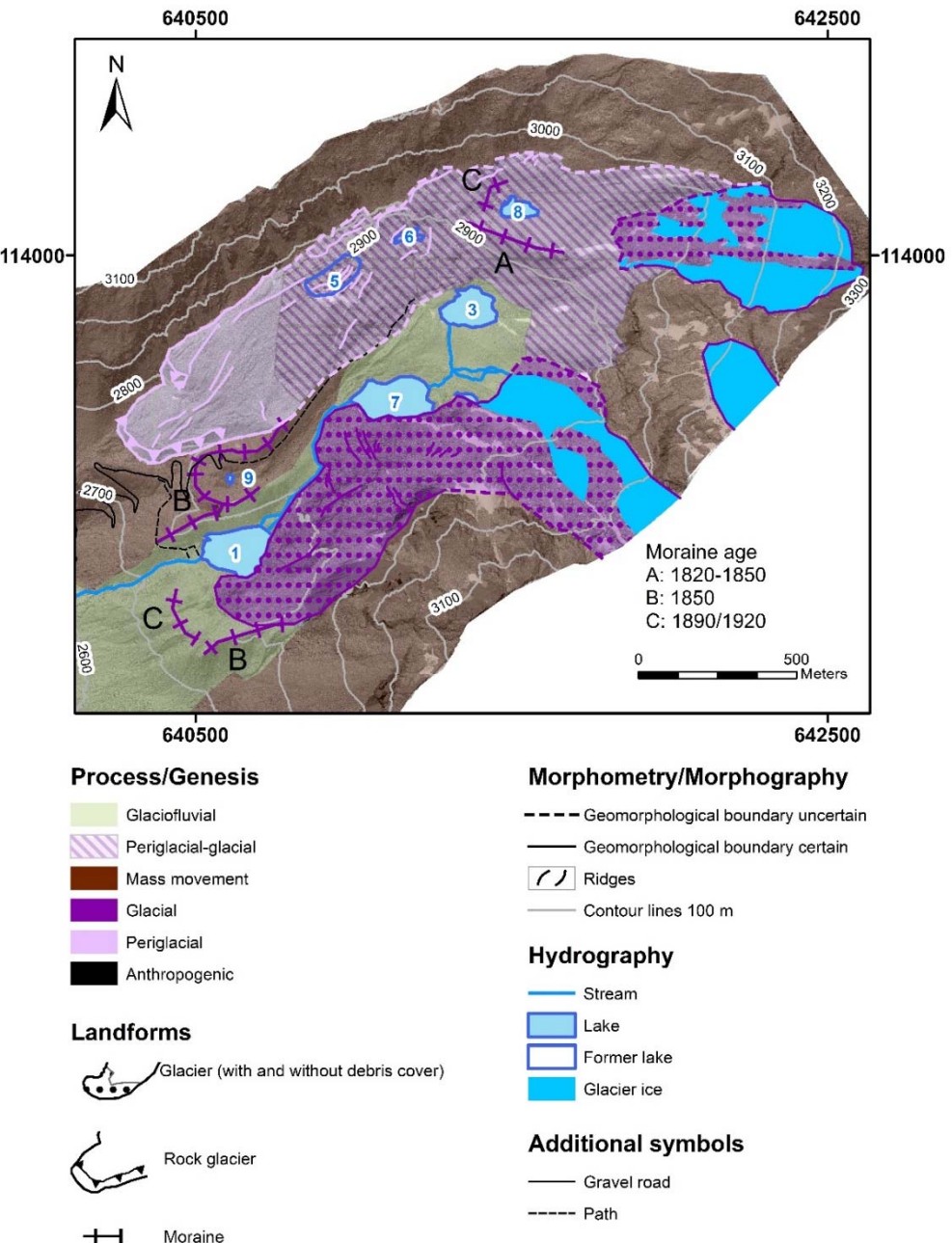

**Process/Genesis**

- Glaciofluvial
- Periglacial-glacial
- Mass movement
- Glacial
- Periglacial
- Anthropogenic

**Landforms**

- Glacier (with and without debris cover)
- Rock glacier
- Moraine

**Morphometry/Morphography**

- Geomorphological boundary uncertain
- Geomorphological boundary certain
- Ridges
- Contour lines 100 m

**Hydrography**

- Stream
- Lake
- Former lake
- Glacier ice

**Additional symbols**

- Gravel road
- Path

225

**Figure 4: Geomorphological map of the Gruben site. The moraine ages are taken from Whalley, 1979 (age A) and Haeberli et al., 1979 (ages B and C). The numbering of lakes 1 to 9 follows the historical sequence of lake formation and denomination; lakes 2 and**

**4 are not anymore present. The mapping is conducted on the orthophoto and DTM as of 2016 (© Swisstopo). Axes are labeled with Swiss coordinates (in meters; CH 1903).**

## 3 Geomatics: Data & methodology

Geometry changes of Gruben rock glacier and the debris-covered part of Gruben glacier were quantified by the application of digital photogrammetry based on digitized or digital aerial photographies. In order to continue the existing kinematic monitoring series (1970 - 1995) at the Gruben site (Kääb et al., 1997; Kääb, 2001)), large-scale aerial images of the years 1994, 2000, 2006, 2010, and 2016 (Swisstopo) were selected for this study. Image orientation, automatic generation of Digital Elevation Models (DTMs) and digital orthoprojection were performed within the digital photogrammetry software SocetSet (BAE Systems, UK). DTMs with 1 m spacing and orthophotos with 0.32 m ground resolution were compiled for the 1994, 2000, 2006, and 2010 imagery, while the 2016 data from aerial digital sensor (ADS) were already processed to DTMs and orthoimages. The accuracy of the DTMs is estimated to lie within the range of 1 - 3 pixels for moderate high mountain topography (Kääb, 2005). The DTMs were co-registered according to the procedure by Nuth and Kääb (2011). For the quantification of vertical changes, the DTMs of the respective years were differenced within ArcGIS. Thus, loss of excess permafrost ice and glacier ice, as well as the topographic expressions of mass transfer can be quantified. The horizontal velocities were calculated using standard digital cross-correlation as implemented in the software CIAS (Correlation Image Analysis Software; Kääb and Vollmer, 2000; Heid and Kääb, 2012). The error (root mean square) of single displacement measurements is about +/- 0.3 – 0.4 m or, in the case of a 5-year interval between two photo missions, about +/- 0.06-0.08 m/a and the error of the vertical changes is estimated to +/- 1 m or, +/- 0.17 – 0.2 m/a (Kääb et al. 1997; Brunner, 2020). If a large number of measurements are analyzed in combination (as e.g. in Figure 8), the statement is an order of magnitude more accurate (+/- 0.006 – 0.008 m/a). Detailed information on raw data, processing steps, resampling methods and data accuracy is given in (Kääb, 2005; Roer et al., 2005a; Brunner, 2020). The horizontal velocities shown in this study are given for values above a maximum correlation coefficient determined as the 30% quantile, as detailed in Brunner (2020).

In addition, repeated GNSS (Global Navigation Satellite System) measurements were started on the rock glacier in 2012 (Barboux et al., 2015; cf. Beutel et al., 2021, for additional measurements). At 46 locations, measurements have been taken twice a year around 1st of July and 1st of October to estimate seasonal as well as annual and decadal changes. The measurement points are distributed over different zones in the periglacial and the glacier-affected part of the rock glacier (Fig. 6). According to the recurrent measurement of four stable control points outside the rock glacier, the accuracy (standard deviation) in positioning is 0.8 cm in horizontal coordinates and 1.0 cm in elevation. The uncertainty in horizontal velocity as well as in vertical displacement rate, without taking into account any tilting or specific movement of the marked boulder, is about 2 cm/a over a year, but rises to 3 cm/a when the velocity is computed over the 9 winter months and 9 cm/a over the three summer months. Further, permanent GNSS data are available for the Gruben rock glacier from 2012 onwards. The fixed station provides high-resolution data (seasonal to sub-seasonal) on surface deformation (Beutel et al., 2021). Both GNSS data sets can be compared with the multi-annual velocities obtained from cross-correlating repeat orthophotos of the rock glacier.

## 4 Results

### 4.1 Gruben rock glacier – *periglacial part*

For the period 1970 – 1995, Kääb et al. (1997) showed horizontal velocities of several decimetres per year for the central part of the rock glacier (= upper part of the periglacial part) and maximum values of about 1m/a directly above the rock glacier front. The vector field depicted a uniform pattern over the years and a sharp velocity increase as the rock glacier creeps over a bedrock riegel about 250 m above the front (see Fig. 1) indicated by (unpublished) seismic refraction soundings (Kääb, 2005). In the following years (1994 - 2016), as investigated in the present study, the overall pattern of the vector field remained similar (Fig. 5). Towards the rock glacier lateral margins the horizontal velocities are below 0.1 m/a and mostly in the range of measurement uncertainty. Especially the orographic left part of the rock glacier seems to be hardly active. As in the previous study, velocities between 0.2 – 0.4 m/a occur in the central part of the rock glacier and higher rates of 0.7 – 1 m/a at and below the bedrock riegel (Fig. 1). Mean horizontal velocities of the periglacial part of the rock glacier for the period 1970 to 2016 are given in Fig. 8a and indicate only little changes (about 0.3 m/a). The in-situ measurements in the last decade (2012 - 2020) give similar results and show in addition that the rock glacier surface near the front (points 040 - 043) is moving about 25% slower than the tongue behind (points 032 - 039). Only small changes in horizontal velocities are observed over time with an almost constant trend around which limited seasonal and interannual variations occur, whereas the very terminal part of the rock glacier above its front tends to decelerate by about 2.5 cm/a (Fig. 7).

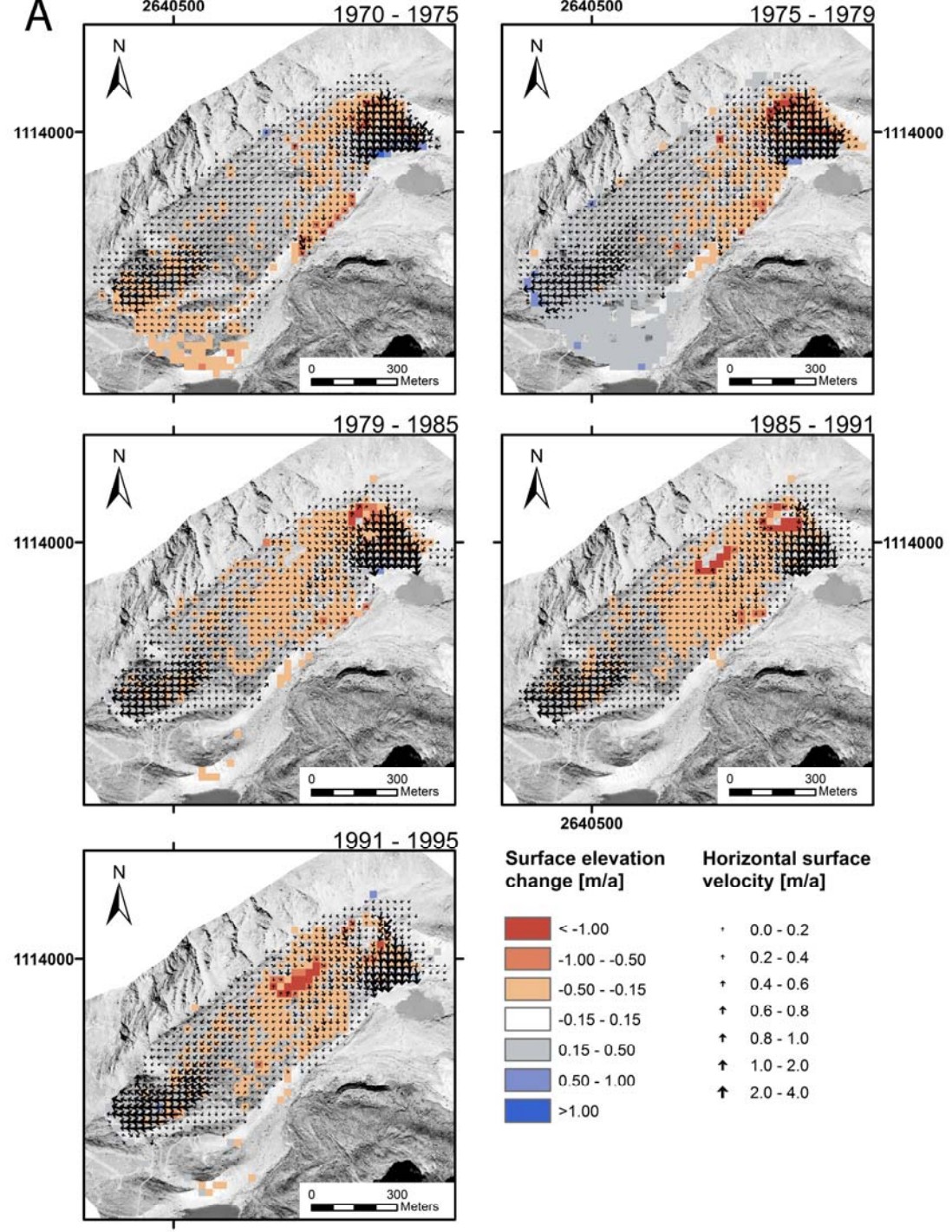

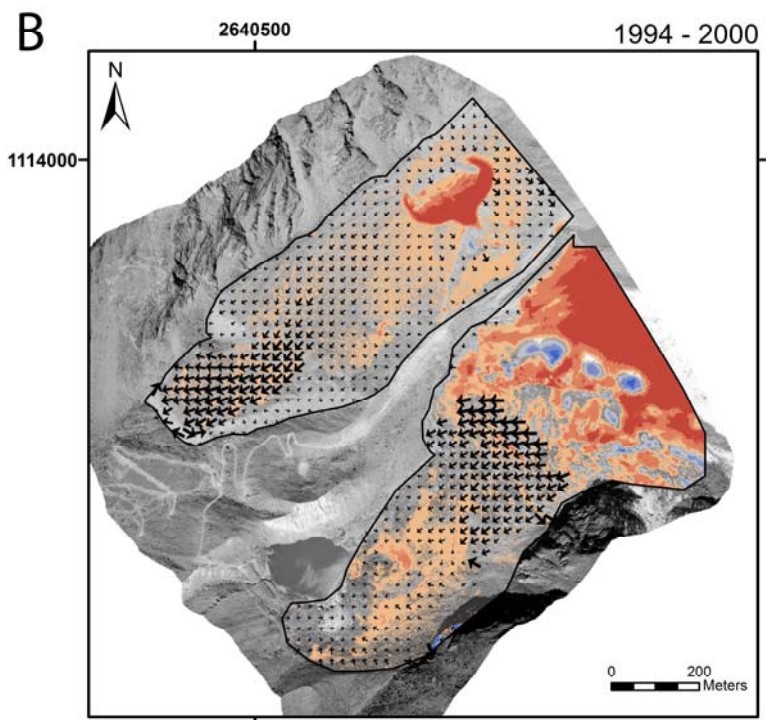

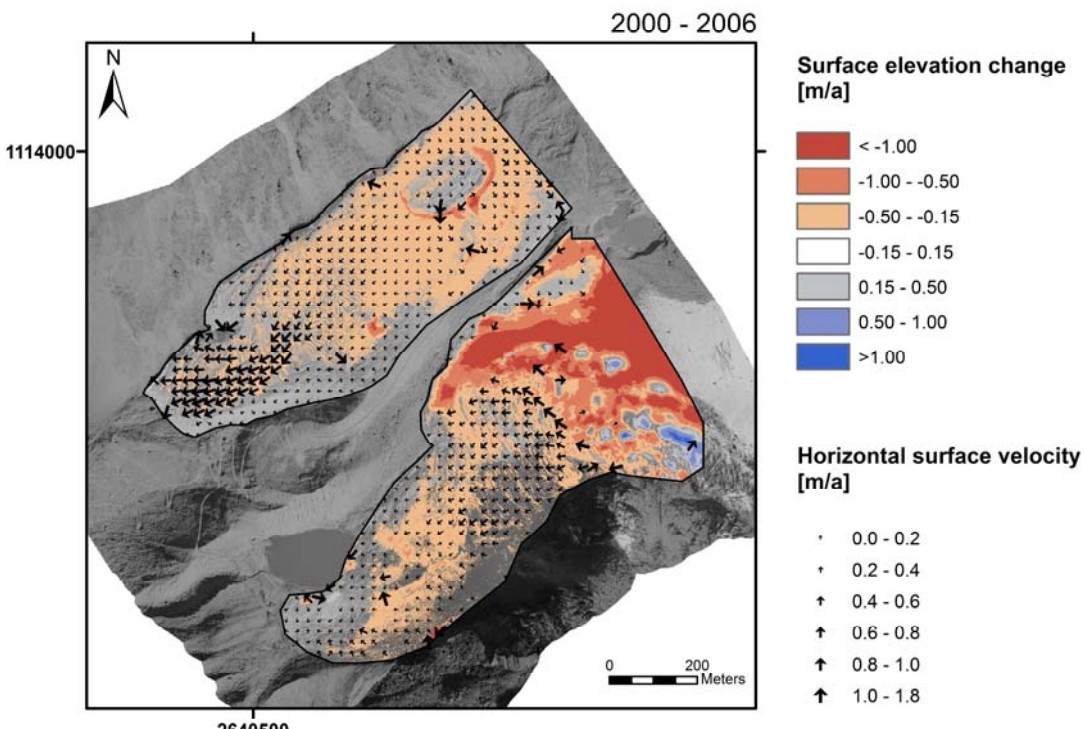

2000 - 2006

**Surface elevation change [m/a]**

- < -1.00
- -1.00 - -0.50
- -0.50 - -0.15
- -0.15 - 0.15
- 0.15 - 0.50
- 0.50 - 1.00
- >1.00

**Horizontal surface velocity [m/a]**

- 0.0 - 0.2
- 0.2 - 0.4
- 0.4 - 0.6
- 0.6 - 0.8
- 0.8 - 1.0
- 1.0 - 1.8

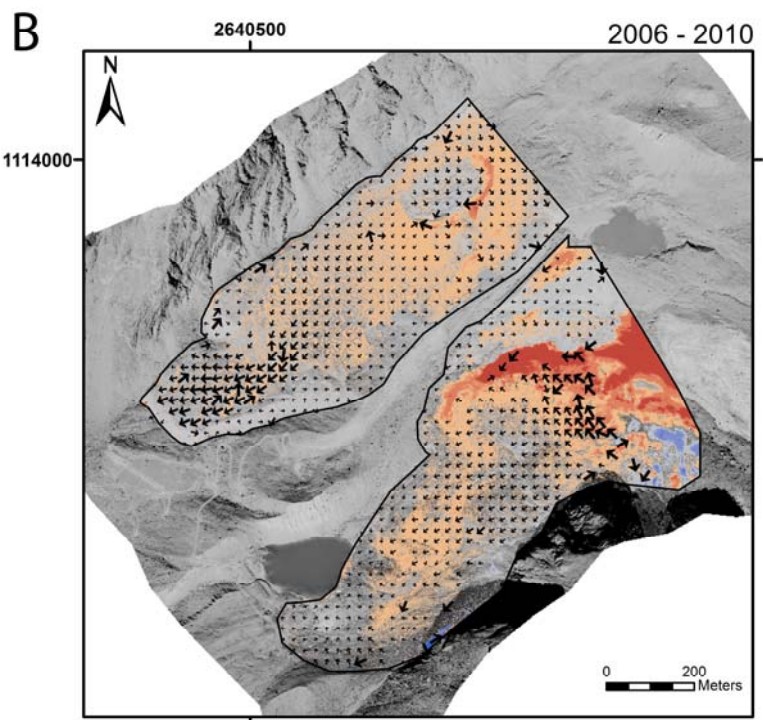

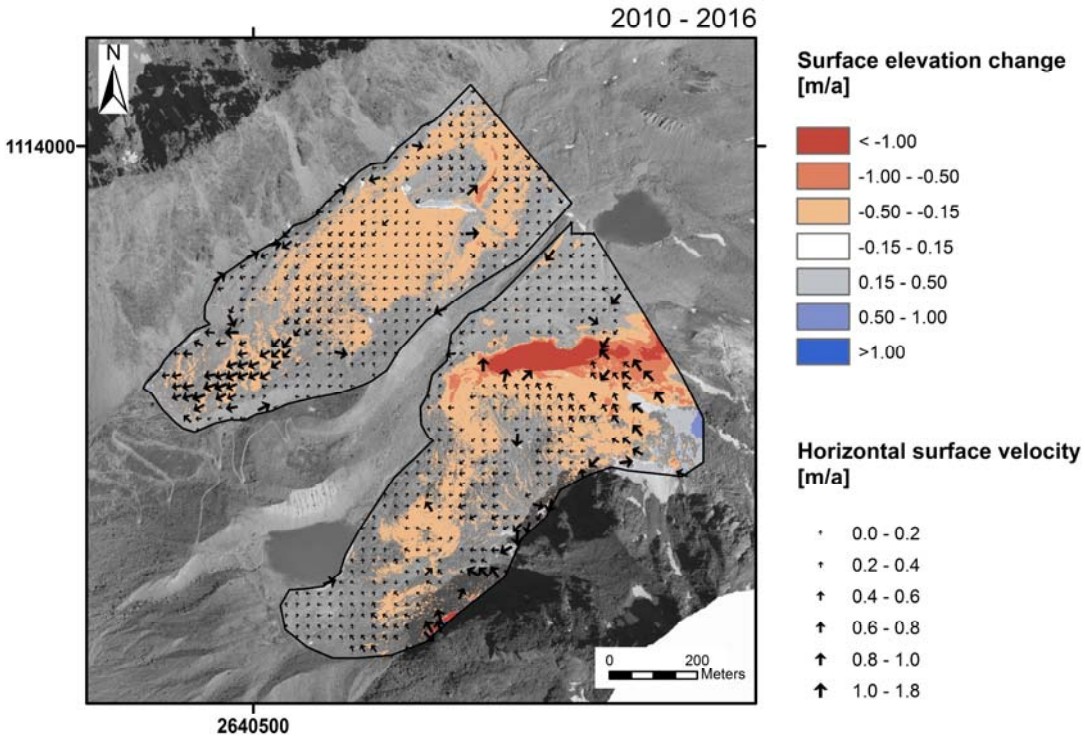

**Surface elevation change [m/a]**

| | |
|---|---|
| 🟥 | < -1.00 |
| 🟧 | -1.00 - -0.50 |
| 🟨 | -0.50 - -0.15 |
| ⬜ | -0.15 - 0.15 |
| ⬜ | 0.15 - 0.50 |
| 🟦 | 0.50 - 1.00 |
| 🟦 | >1.00 |

**Horizontal surface velocity [m/a]**

| | |
|---|---|
| · | 0.0 - 0.2 |
| ⸱ | 0.2 - 0.4 |
| ↑ | 0.4 - 0.6 |
| ↑ | 0.6 - 0.8 |
| ↑ | 0.8 - 1.0 |
| ↑ | 1.0 - 1.8 |

**Figure 5: Overlay of the horizontal surface velocities (vectors) and the colour-coded changes in surface elevation; 5A for the years 1970 to 1975, 1975 to 1979, 1979 to 1985, 1985 to 1991, and 1991 to 1995 for the rock glacier only (from Kääb, 1997), 5B for the years for 1994 to 2000, 2000 to 2006, 2006 to 2010, and 2010 to 2016 for the rock glacier and the debris-covered glacier tongue (for each observation period the later orthophoto is used as background; 2000, 2006, 2010, and 2016, respectively). Axes are labeled with Swiss coordinates (in meters; CH 1903).**

The vertical changes in the central part of the rock glacier reflect surface lowering (-0.1 to -0.5 m/a) between 1970 and 1995, whereas the lower tongue seems to remain almost constant in thickness (Kääb, 2005). In the period 1994 – 2016 the vertical changes show the same pattern and are in about the same range of -0.1 to -0.5 m/a (in total between -2 and -10 m). The lower part of the rock glacier shows little vertical changes (Fig. 5). Mean vertical changes for the period 1970 to 2016 are given in
Fig. 8a and indicate slow and nearly constant surface lowering.
The GNSS measurements of the period 2012 - 2020 show that the trajectories of the marked boulders at the rock glacier surface are all 10 to 30° inclined, what is close, but in most places somewhat steeper than the surface topography (Fig. 7). The trajectories are often steeper in summertime, indicating ice-melt induced subsidence between 0 and -0.3 m/a. The absence of a relative height gain caused by the compression related to the slow-down of the creep velocity in the last tens of meters above
295 the front indicates the occurrence of an ice-melt induced subsidence, which can reach here up to -0.5 m/a. The data of the permanent GNSS, situated between the bedrock riegel and the rock glacier front, fit well with the data of the photogrammetry and the GNSS and show values between 0.9 – 1.1 m/a.

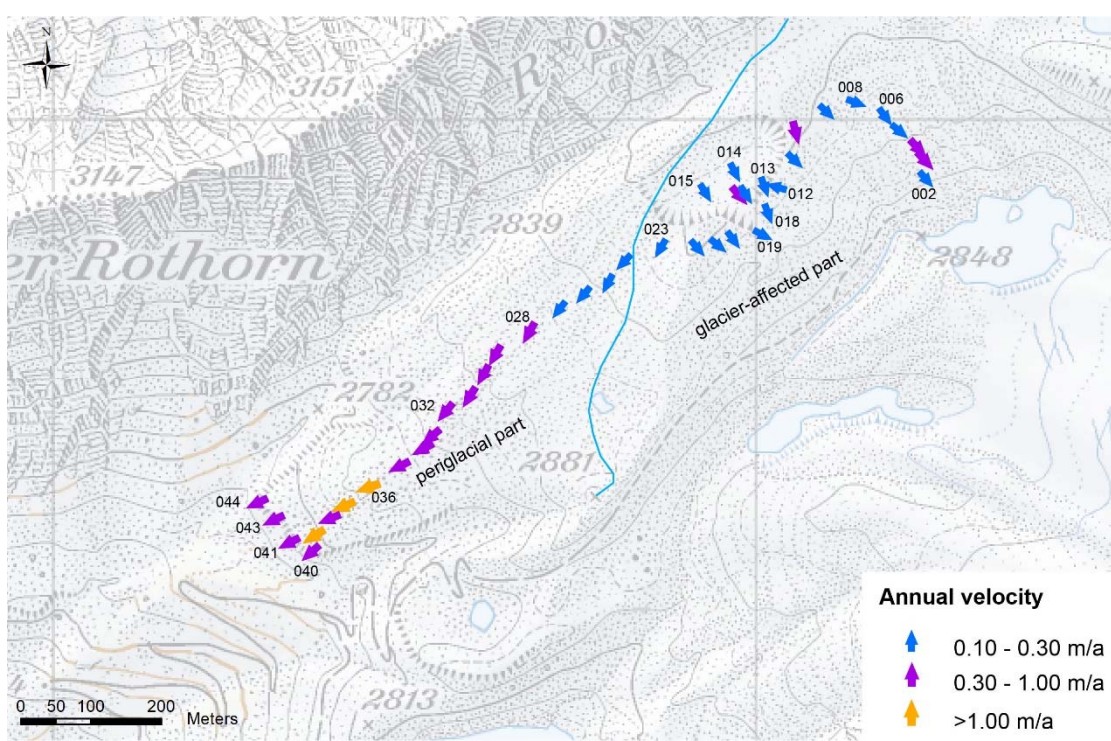

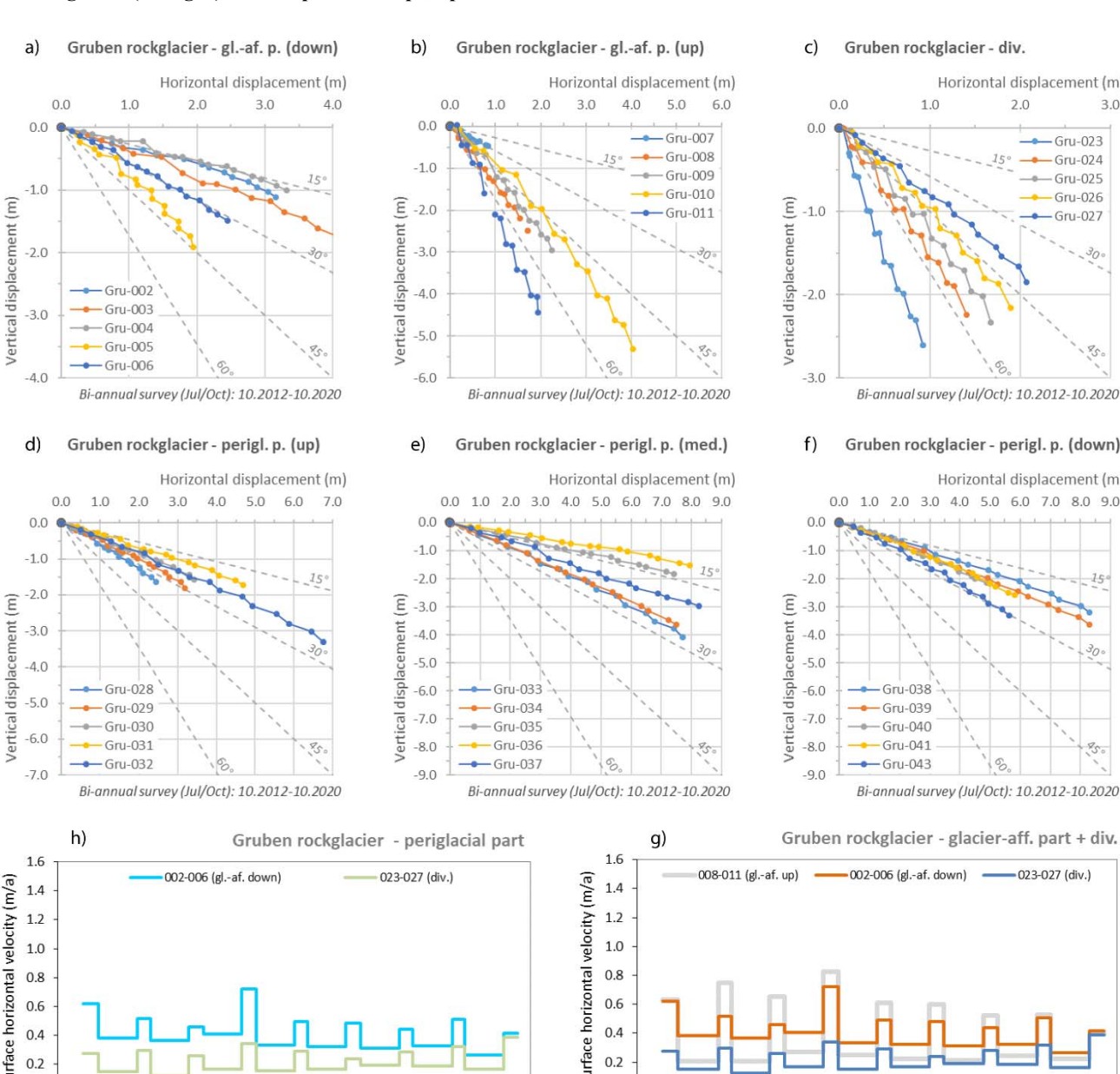

**Figure 6: Overview on horizontal flow fields on Gruben rock glacier, measured by GNSS for the year 2019/2020. Numbers are referring to GNSS measurement items. The blue line separates approximately between the glacier-affected and periglacial parts of the rock glacier (cf. Fig. 1). Basemap © Swisstopo, updated 2005.**

**Figure 7: GNSS-based kinematic behaviour in both glacier-affected (gl.-af.) and periglacial (perigl.) parts of the Gruben rock glacier as well as in their transition zone (div.). a) - f): Horizontal versus vertical displacement of selected items. g) - h): Seasonal horizontal**

**velocities in specific sections. The numbers or identifiers are referring to the GNSS-surveyed items; location in Fig. 6.**

## 4.2 Gruben rock glacier – *glacier-affected part*

The former contact zone between the polythermal Gruben glacier and the periglacial permafrost of Gruben rock glacier is here called glacier-affected part. The cold glacier margin most likely pushed the permafrost upslope during its extended Little Ice Age stages, while the changes in slope and stress fields induced by the retreating glacier margin more recently have caused permafrost to creep back into the direction of the topographic depression formerly filled with the glacier tongue. As detailed by Kääb et al. (1997), the horizontal velocities between 1970 and 1995 in this part of the glacier - rock glacier contact were significantly higher than in the periglacial part and reached up to several meters per year near the glacier-dammed lake 3. These high creep rates were accompanied by strong vertical changes resulting from melting of dead ice and the formation and growth of thermokarst lakes. In the former contact zone of the glacier and the rock glacier (now the proglacial area), the retreat of the glacier caused a strong but decelerating surface lowering over the 25-years period (Kääb et al., 1997; see also Fig. 8).

In the following years (1994 - 2016), the mean horizontal velocities in the glacier-affected part of the rock glacier remained almost stable at about 0.2 m/a (see mean horizontal velocities in Fig. 8b). The flow is still directed towards the debris-free part of the glacier. The vertical changes show the highest mean value of -0.3 m/a between 1994 and 2000, followed by a decreasing trend of vertical losses reaching a mean value of -0.14 m/a. The area of the thermokarst lake 5 (Fig. 1) indicates high vertical losses of up to 25 m between 1994 - 2016 at the southern lake outlet related to the artificial draining of the lake and the collapse of its former lake margins (Fig. 5). In addition, vertical changes related to the engineering works carried out in 1995 (road construction and trench digging) are clearly visible in the comparison of the 1994 and 2000 DTMs. Due to the former lake surface and the collapse of lake margins, accompanied by a loss in visual coherence, tracking of blocks and quantification of horizontal displacements could not be conducted in this part of the landform (especially between 1994 and 2000).

The GNSS measurements of the period 2012 - 2020 show that most of the area, except the transition zone toward the periglacial part, is still back-creeping at a mean rate which has tended to decrease from about 0.4 to 0.3 m/a. The trajectories of the marked surface boulders are 40° to more than 70° inclined in the uppermost glacier-affected part (thermokarst lake area) and in the transition zone toward the periglacial part. The steepness of the trajectories is essentially caused by a strong ice-melt induced subsidence in summertime combined to a low rate of horizontal movement. The subsidence is ranging from 0.25 to more than 0.5 m/a, thus showing slightly higher values than the photogrammetric analysis. Trajectories in the downstream part of the back-creeping zone (pts. 002 - 004) are 15 - 20° inclined (Fig. 7), which is much closer to the topographic slope angle. Steeper trajectories in summertime cannot be evidenced, showing the absence of any significant ice loss at depth.

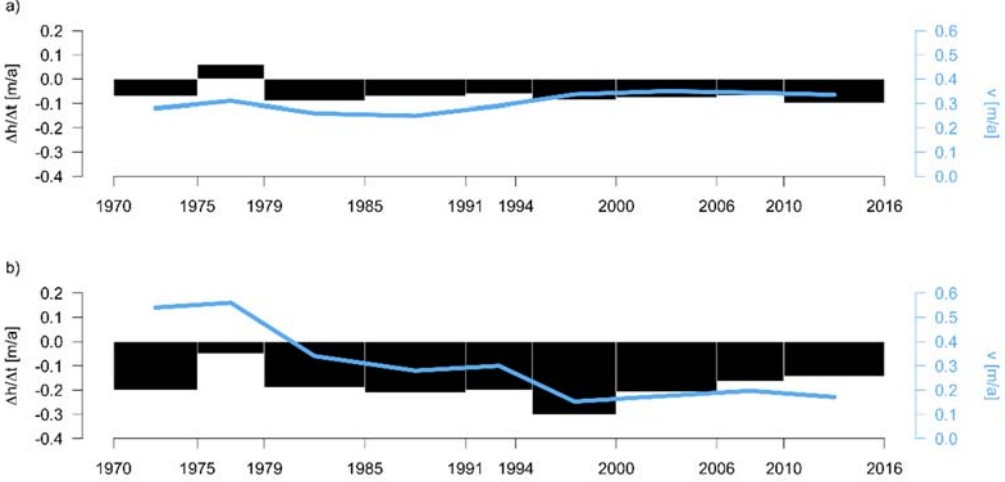

**Figure 8: Mean vertical changes (black columns) and horizontal velocities (blue line) of the periglacial part of the rock glacier (a) and the glacier-affected part of the rock glacier (b) from 1970 to 2016. The values of the period 1970 to 1995 result from Kääb et al., 1997.**

### 4.3 Gruben glacier – *debris-covered tongue*

In prior studies, the research focussed on the debris-free tongue of Gruben glacier, on the rock glacier and on the development of lakes. Kääb (2001) compiled measurements on the debris-covered tongue on Gruben glacier. In the study presented here, detailed horizontal velocities were not only quantified for the rock glacier, but also for the entire debris-covered part of Gruben glacier, in order to compare spatio-temporal patterns of the different landforms and to analyse the ongoing processes. As it becomes apparent in Fig. 1, the debris-covered glacier tongue has the same orientation as the rock glacier as well as a similar altitudinal range but receives much more shadow from the Inner Rothorn. The surface roughness of both landforms is comparable, with a high percentage of large blocks (> 1 m in diameter). In relation to the sediment input by rock falls, the boulders appear to be less sorted on the debris-covered glaciers than on the rock glacier. The surface topography of the debris-covered glacier tongue shows somewhat subdued ridges perpendicular to the flow lines, which are probably related to a combination of compressing flow, thrusting, differential ablation and downwasting of the glacier, which consists here of cold ice and is likely frozen to the subglacial sediments. The surface structure strongly contrasts with the striking longitudinal structures of the rock glacier with its predominantly extending flow (Fig. 1; cf. also Kääb and Weber, 2004). In addition, the terminal part of the debris-covered glacier constitutes a diffuse transition to its foreland at lake 1 and does not exhibit the characteristic sorting of material as it is the case for the striking over-steepened fronts produced by creeping masses of frozen talus/debris of actively advancing rock glaciers.

For the debris-covered tongue, a general decrease in velocity from 0.27 m/a (1994 - 2000) to 0.17 m/a (2010 - 2016) is described by the median horizontal surface displacements (Brunner, 2020). In addition, the vector field depicts a division into two parts. The lower part shows surface velocities of 0.3 m/a decreasing towards zero at the lower ice margin and vectors show

a whirl-like orientation, indicating stagnation of flow. The upper part shows much higher but decelerating velocities, and has a flow field essentially following the main slope. Vertical changes on the debris-covered tongue lie in the range of -0.1 to -0.8 m/a apart from the uppermost part which is in close connection to the main body of the glacier, where mean surface lowering amounts up to 17 m between 1994 and 2016 (about -0.77 m/a) (Brunner, 2020). Here, at the glacier margin, lake 7 with a pronounced ice cliff heavily affected by thermokarst processes recently formed.

### 4.4 Gruben glacier – *debris-free tongue*

On the ice tongue of Gruben glacier, only a few horizontal velocities were quantified in this study, due to unsuitable temporal resolution of the aerial images with respect to glacier flow. But glacier retreat of about 370 m (on average nearly 20 m per year) is quantified for the measurement period 1994 - 2016. In addition, surface changes measured from DTM comparison indicate a lowering of up to 47 m at the glacier margin in this period (on average about 2 m per year), much more than found for the 1970s – 1990s (Kääb, 2001).

This quantification allows for a comparison with earlier studies measuring and modeling ice thickness and analyzing the glacier bed, such as Haeberli and Fisch (1984), who applied a combination of thermal drilling and electrical resistivity soundings of subglacial material, documenting an ice thickness between 25 m and 80 m along a 400-m profile over the glacier tongue. An about 150 m thick layer of unconsolidated sediments occurs in the glacier bed underneath the tongue (Haeberli and Fisch, 1984). Comparing the ice thickness of about 35 - 40 m as measured in 1979 at the former glacier tongue with the same area (now ice free) and the ice loss of more than 40 m between 1994 and 2016 indicates a good coincidence.

## 5 Discussion

### 5.1 Periglacial processes

The periglacial part of Gruben rock glacier shows a coherent flow field, a thickness remaining nearly constant in time and a steadily advancing over-steepened front, pointing to continued creep of ice-rich permafrost with its pronounced thermal inertia (Kääb et al., 1997; Kääb, 2005). Consistent movement rates are derived from photogrammetry and GNSS data, which fit well with the data of a permanent GNSS, situated between the bedrock riegel and the rock glacier front (Beutel et al., 2021). With annual velocities of 0.5 - 1.0 m/a it lies within the range of typical rates for permafrost creep in rock glaciers of Switzerland (Delaloye et al., 2010). The advance rate of the front is significantly smaller than the surface velocities (3 m versus 17 m between 1970 and 1995; Kääb et al., 1997), which also resembles a typical pattern and links to a probable shallower shear horizon and/or melting processes at the front. The spatial pattern of its flow field essentially remained constant with only small temporal variations in the last two decades. Characteristic rates of surface lowering are in the range of centimeters to few decimeters per year, remain relatively constant over time (Fig. 5 and 8) and are comparable to observations on other rock glaciers (Bodin et al., 2009; Fey and Krainer, 2020; Cusicanqui et al., 2021; Kääb et al., 2021). The annual in-situ measurements allow for an interpretation and distinction of different processes causing vertical changes, such as

extension/compression, melt-induced subsidence or topographic influences (Arenson et al., 2002; Lambiel and Delaloye, 2004). The overall extending flow regime of the rock glacier is generating a subsidence of a few centimeters along most of the GNSS longitudinal profile, reaching up to 20 cm/a above the bedrock riegel, whereas a compression heave of more than 20 cm/y should occur close to the front. This rate is proportional to spatial velocity contrasts and, because the flow field is constant over time and the velocity not dramatically changing over the seasons, it is not expected to vary much over the year. The in-situ measurements, however, indicate higher rates of vertical motion during the summer months, e.g. in the hot summer of 2015 (Fig. 7), which seem to evidence thaw subsidence in large sections of the periglacial part of the rock glacier. Part of the vertical change must be related to the predominantly extending flow and related mass transfer. In reaction to increasing air and ground temperatures, pronounced flow acceleration and surface lowering or even flow destabilization (Roer et al., 2008) have been observed at a good number of rock glaciers in the Alps (Delaloye et al., 2010; PERMOS, 2020; Fey and Krainer, 2020, Cusicanqui et al., 2021; Kaufmann et al. 2021) and other mountain ranges (Darrow et al., 2016; Eriksen et al., 2018; Kääb et al., 2021). The most plausible reason is that progressive permafrost warming causes subsurface ice to become softer, to contain higher amounts of unfrozen water and to increase the hydraulic permeability of the ice-rock mixtures (Kääb et al., 2007; Haeberli et al., 2013; Cicoira et al., 2019). The steady behavior of Gruben rock glacier over the last 50 years is in contrast with such warming-induced acceleration of viscous creep in perennially frozen debris but resembles long-term developments in the Tien Shan (Kääb et al., 2021). There, marked acceleration is observed for large rock glaciers with the exception of a glacier-affected case (Gorodetsky), where glacier retreat and dead-ice melting may decouple landforms of viscous permafrost creep from debris supply and induce changes in the stress field (unloading). In the case of Gruben, the retreating orographic right part of the largely debris-free glacier never supplied much debris as clearly recognizable from Fig. 1 and from old maps publicly available at Swisstopo (2022; cf. supplementary material S6). Rather, the retreat of the glacier margin induced a south-orientated backflow of the upper rock glacier part towards the now exposed topographic depression (overdeepening) of the former glacier bed, away from the generally southwestern flow direction towards the rock glacier. Other possible influences remain highly speculative and require further investigation or modelling efforts. Thinning of the rock glacier, which can reach up to 3 meters per decade in the transition area between the periglacial and glacier-affected parts, is gradually, but clearly decreasing the stress at the shear horizon. High ice contents could, in principle, dampen the temperature penetration with depth due to latent heat exchange caused by thaw at depth and hence prevent the warming from reaching shear horizons at depth, where most of the deformation takes place (Kääb et al., 2007). In addition, a high ice content can influence the hydrological system, and dampen the water flow and hydraulic and thermal connectivity within the rock glacier, thus making the landform insensitive to seasonal and interannual temperature forcing. The information from the geophysical soundings, however, provides no evidence of extraordinary high ice contents or even large bodies of massive ice such as they have been documented from the rock glacier Murtèl (Hoelzle et al., 2002; PERMOS, 2020). The role of the bedrock riegel, which regulates and limits flow depth in the lower rock glacier part, is also difficult to judge. Ultimately, continued atmospheric temperature rise will unavoidably cause progressive permafrost degradation and thaw during the upcoming decades and centuries, accompanied by deceleration and inactivation of rock glaciers, as the melting of excess subsurface ice will lead to a loss of cohesion, an increase

in internal friction, and ultimately slow down and stop movement through viscous flow. Intermittent flow acceleration could occur in a first stage. Continued thermal and kinematic monitoring at this well-documented site and the detailed analysis of high-resolution data from the permanent GNSS (Beutel et al., 2021), can shed more light on the short-term and long-term changes, as well as on involved processes.

The periglacial part of the rock glacier shows a striking coincidence of modern flowlines and (longitudinal) structures from long-term cumulative deformation. In strong contrast to this, the former glacier-affected part exhibits a rather extreme divergence between modern flow directions and surface structures from long-term cumulative deformation (Fig. 1, 5, 6). Due to glacier vanishing and unloading with related topographic changes since the Little Ice Age, this part experienced a marked reorientation of the stress and flow fields. During the entire observation and probably much longer, the strikingly convex

landform "rock glacier" underwent slow cumulative deformation and advance through viscous creep of the frozen talus material but essentially kept the same appearance. Characteristic features are perfectly preserved such as (i) an "organized" surface structure strikingly reflecting a coherent field of cohesive viscous creep with large-scale stress transmission, here primarily under a regime of longitudinal as well as transversal extension, and (ii) an over-steepened, unstable front of the continuously advancing frozen mass where fresh material from the inner parts of the rock glacier is continuously being

exposed. In sharp contrast to this persistent landform evolution, clear signs of down-wasting and collapse characterize the debris-covered, cold part of the Gruben glacier tongue. When directly comparing orthoimages from the 1970s to those from 2016 one can easily track individual rocks or patterns of rocks over time on most areas of periglacial rock glacier part, something that is  more difficult on the glacier-affected part, and on the debris-covered glacier tongue only possible over smaller disconnected areas.

**5.2 Glacial and proglacial processes**

The changes on the debris-free and the debris-covered part of Gruben glacier indicate typical processes in reaction to the changing climate with increasing temperatures (Zemp et al., 2015). The largely debris-free part of Gruben glacier has been thinning by tens of meters and retreating by hundreds of meters. In analogy to Alpine glaciers in general, characteristic thinning rates during the observation period (1970 - 2016) are several decimeters per year (cf. Haeberli et al., 2001; Sommer et al.,

2020). During its maximum LIA advance, the glacier with its cold margins must have exerted compressive stresses to the upper part of the rock glacier. This may have pushed ice-rich frozen debris upslope, towards the talus at the foot of the Outer Rothorn crest. Flow trajectories in this former contact zone are presently in a near-opposite direction, i.e. from the talus towards the now ice-free topographic depression (overdeepening) around lake 3. This is likely a consequence of the approximately 180° change in slope and corresponding stress re-orientation induced by glacier retreat. Where the glacier disappeared,

completely new landscapes started forming. The occurrence of fluted moraines in the northern part of the cirque serves as a testimony that the polythermal glaciers had been partially warm-based and largely debris-free. The debris-covered, cold orographic left part of the Gruben glacier shows similar developments as its debris-free part, but strongly lagged and partly hidden due to the protecting cover of debris influencing near-surface heat exchange. Similar to the contact zone between the

polythermal Gruben glacier and the permafrost of Gruben rock glacier it mainly shows intermediate thinning rates. The uppermost part of the debris-covered glacier tongue, which is still in contact with the active glacier, shows strong signs of down-wasting. Here, the velocities still indicate a coherent flow field with decelerating velocities over the observation period, while the lower part of the debris-covered tongue shows a marked decrease in velocity towards stagnation. This documents the retrogressive influence of the retreating and downwasting glacier. Despite somewhat vague indications of compression in the debris cover (rather weakly pronounced transverse ridges), the surface structure remains predominantly chaotic and the margins of the landform remain diffuse to such a degree that judging the exact position of the "ice margin" becomes hardly possible without the use of additional data, such as radar imagery (Fischer et al., 2014). Even under similar thermal conditions (negative annual surface temperatures), the difference between the rock glacier and the debris-covered glacier is striking.

For both parts of the glacier, the retreating and down-wasting ice is the driving process exposing and forming a new landscape. The proglacial area is typically dominated by gravitational and fluvial processes, but is still conditioned by the former glaciation. This adjustment from glacial to non-glacial conditions is formulated by the paraglacial concept and describes a transition towards new equilibrium conditions (Ballantyne, 2002; Curry et al., 2006; Slaymaker, 2011). There is a scientific need to observe and quantify geomorphological changes, characteristic successions, as well as long-term trends in order to better understand the evolution of proglacial systems during this transition phase (Carrivick and Heckmann, 2017). In our study we provide insights into glacial, periglacial and paraglacial dynamics over a period of 50 years. One interesting feature of the glacial system is the delayed response of the debris-covered tongue to the atmospheric changes (several decades) and the related deceleration of ongoing geomorphological processes.

The ice in the former contact zone ("glacier-affected part") between the polythermal Gruben glacier and the permafrost of Gruben rock glacier generally shows intermediate thinning rates. Locally, vertical changes are higher in relation to the artificial emptying of the thermokarst lake and the melting of glacier remains. Horizontal velocities show a marked decrease with time. These observations indicate the fading interaction between glacial and periglacial processes and the rapid stabilization of the ground, as typically found in proglacial systems (Carrivick and Heckmann, 2017). Assuming characteristic mean thicknesses of several tens of meters for both, the glacier and the ice-rich frozen ground, the time scale for complete ice loss as derived from such quantitative rates of subsidence and mass loss is decades to centuries for the clean glacier while it is at least an order of magnitude longer for the rock glacier ice. Subsurface ice in perennially frozen rock glaciers will continue to exist under conditions of continued warming when most surface ice in glaciers will already have disappeared (Haeberli et al., 2017). The creeping permafrost body is thereby most likely far out of thermal equilibrium already today and in its lower part could indeed now be approaching near-isothermal thawing temperatures at depth.

## 5.3 Future perspectives and related hazard aspects

Since the earlier 20[th] century, the complex evolution of glaciers and permafrost in the Gruben cirque has led to the development of an interconnected system of smaller but nevertheless hazardous lakes. Engineering mitigation work as combined with thorough scientific investigations and monitoring enabled successful prevention of further damaging incidents after the

outburst floods in 1968 and 1970. The situation should nevertheless be carefully kept under observation. In general terms, with the accelerated glacier shrinking and the related destabilization of adjacent slopes, catastrophic mass flows can be triggered (Huggel et al., 2005; Deline et al., 2021; Curry et al., 2006). One challenging aspect thereby concerns the obvious destabilization of the shady rock walls at the Inner Rothorn with strong effects from glacial de-buttressing and permafrost degradation. Activation of rock falls has been evident during field work for many years already and future large-volume events cannot be safely excluded. Such events could in principle have the potential to suddenly and entirely displace water bodies with the dimensions of the lakes at the site (Haeberli et al., 2017). The early transformation of lake 1 into a flood retention basin with a capacity of about 100,000 m³ of water has so far provided protection against floods in the valley but this retention basin should better not be full of water in case of a large rock avalanche potentially reaching it. Since both, the glacial as well as the periglacial systems are highly dynamic, new lakes may form in future at the immediate foot of the destabilizing rock wall; e.g. between the debris covered part of the glacier and the active glacier tongue assuming an ongoing retreat of the glacier (cf. Frey et al., 2010). In comparison with such rapid developments, rock glaciers with their creeping frozen materials constitute persistent elements of cold mountain landscapes. Whether current and future warming accelerates or stops, or accelerates and then stops, rock glacier creep remains to be observed. Continued monitoring of lake developments, further down-wasting and retreat of glacier ice, and of permafrost creep in the rock glacier tongue is conducted by national (PERMOS, GLAMOS) and local authorities and will allow for assessments (cf. GAPHAZ, 2017) of future hazard situations, if applicable.

**6 Conclusions**

Long-term monitoring of glacial and periglacial processes, materials and landforms at the Gruben site using modern geomatics methods opens unique insights into past and ongoing climate-related dynamics of complex ice-related system responses to rapid atmospheric temperature rise and corresponding landform evolution under moderately continental climatic conditions. The measurements document that over the past 50 years:

- the polythermal Gruben glacier retreated significantly as a rapid response to climate change and in accordance with Alpine glaciers in general;
- the cold debris-covered tongue of Gruben glacier degraded gradually from the front backwards and is now containing ice that is far out of equilibrium with the current climate – a situation which is in accordance with other debris-covered glaciers;
- the creeping perennially frozen talus of the Gruben rock glacier showed a steady behavior with a coherent flow field and limited subsidence: as a result of the continued viscous flow it advanced at a rate which remained practically constant during the observation time of five decades;
- the striking lack of an acceleration trend in the viscous creep of Gruben rock glacier due to permafrost warming and softening is not easily explained but may most likely be related to unloading (thinning is still ongoing) and re-oriented surface inclination/stress fields as a consequence of glacier retreat since the LIA in its upper part;

- the geomorphological changes affected the hazard potential at the Gruben site, necessitating human intervention and careful observation combined with regular assessments.

Our analysis of glacial and periglacial processes and landforms and their interactions provides a better understanding of landform dynamics, in particular the current responses of cryospheric landforms to climate forcing. It highlights the various response times in relation to material properties, physical conditions and related process interactions. In addition, it underlines the importance of long-term, integrative monitoring of glacial and periglacial systems, for scientific as well as applied reasons.

## Data availability

GST and GNSS measurements are partly available on the PERMOS Data Portal (www.permos.ch/data.html). All other data are available upon request.

## Author contributions

IGR, WH and AK conceived the study. AB, IGR and AK analyzed the digital elevation models (DEMs) and orthophotos. RD added recent in-situ data. WH contributed all his data and knowledge from earlier investigations. PT supported the processing of digital elevation models and orthophotos. IGR and WH wrote the paper and AB, RD and IGR produced the figures, with support and contributions from all other co-authors.

## Competing interests

The authors declare that they have no conflict of interest.

## Acknowledgements

The DTMs and orthoimages of the Gruben area are based on aerial photographs taken by the swisstopo flight service, provided within the Swiss Permafrost Monitoring Network (PERMOS). Special thanks are due to Andreas Bauder (VAW-ETHZ) and Hermann Bösch (VAW-ETHZ) for access to the aerial images. GST and GNSS measurements are undertaken with the support of PERMOS.

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
