# Peer review of "Glacier-permafrost relations in a high-mountain environment: Five decades of kinematic monitoring at the Gruben site, Swiss Alps"

_The Cryosphere, 2021_

## Author Response (AR1)

**Response to reviews**

Glacier-permafrost relations in a high-mountain environment: Five decades of kinematic monitoring at the Gruben site, Swiss Alps

*Isabelle Gärtner-Roer, Nina Brunner, Reynald Delaloye, Wilfried Haeberli, Andreas Kääb, and Patrick Thee*

We thank all reviewers for their constructive comments. We thoroughly revised the manuscript and the figures based on the feedback. Please find below the individual feedback to all comments.

Best,

Isabelle Gärtner-Roer, with all co-authors

I really enjoyed reading your manuscript. The case of the Gruben site clearly demonstrates the complex dynamics of the glacial, para- and periglacial environments and how their response to climate change varies. There is no simple and unique answer. Long- term monitoring is essential, and your contribution demonstrates the value of such monitoring networks. Overall, the paper is well developed, but the manuscript could benefit from editorial modifications. I added some suggestions in the attached, annotated version of your manuscript. I know that the authors prefer to use the term rockglacier in one word. While I generally agree that using a single term may help to better differentiate rock glaciers from glaciers, the single term usage has not yet been able to assert itself. In fact, it may be counter productive as some readers may say that rock glaciers and rockglaciers are two different landforms. Again, I do understand the rational, but maybe it is more beneficial to use the two-word term. Just a little food for thought.

Dear Lukas, thanks for the constructive comments. We now write "rock glacier" in two words throughout the text. Below you find the answers related to the annotated file:

Line 17: "quite" deleted

Line 19: "obviously" deleted

Line 24: "clean" replaced by "largely debris-free"

Lines 65 and 71: "dangerous" replaced by "hazardous"

Line 81: we prefer to keep "hot-water" as this is more precise than "thermal" ("thermal" also involves electrical heating)

Line 96: "active" added after "perennially frozen"

Line 97: deleted "at around"

Line 101: The term "thermal gradient" was replaced by "environmental lapse rate"

Line 103: time period adapted to «1980 – 2009»

Line 105: this is an estimate, we now write " … around or even slightly below …"

Figure 1: The caption is now rewritten as follows:

Figure 1: Geomorphological description of the Gruben site as shown on an orthophoto from 2017 (Source: SWISSIMAGE, geodata@swisstopo). The blue line indicates the approximate outline of Gruben glacier in the contact zone with the Gruben rock glacier at the time of the Little Ice Age (LIA). The grey line on the rock glacier represents the seismically determined subsurface bedrock riegel. Landforms: A = inactive rock glacier, still frozen; B = actively

creeping, frozen protalus rampart; C = Gruben rock glacier; D = deformation of frozen talus; E = debris-covered tongue of Gruben glacier; F = Gruben glacier; 1, 3, 7 = existing lakes; 5 = former thermokarst lake (lake numbering follows historical scheme in order to keep lake identities constant over time; cf. Figure 4). The inset in the upper right (see full Figure 1 for scale) shows the flow trajectories in frozen debris as determined by aero-photogrammetry with the yellow lines (cf. Figure 14 in Haeberli et al., 1979; and supplementary material S3) and flow trajectories in frozen debris (white dashed lines) which follow the surface structures produced by long-term cumulative deformation of the frozen talus. The blue arrows indicate the estimated flow direction of the LIA glacier as derived from the earliest reliable topographic map (1889; cf. supplementary material S4, S5 and S6).

Figure 2: "G" is corrected into "F" for Gruben glacier. We added " ... (Google Earth 2009; cf. supplementary material S7 and S8 for comparison with other spatial permafrost simulations at the site). Landforms ..."

Lines 124/125: We modified the sentence as follows:

Under such cold-dry conditions, mountain permafrost is widespread (Fig. 2; cf. other model results for the site from FOEN and Kenner et al., 2019 (supplementary material S7 and S8), which are in general agreement with large-scale simulations at 1 km resolution by Obu et al., 2019 (Northern Hemisphere) or Gruber, 2012 (worldwide)).

Line 128: replaced by «centigrades»

Line 135: adjusted into « ... over time. Borehole temperatures at 3m depth close to but below the permafrost table in the upper part of the ..."

Line 138: adjusted to " ... the rock glacier permafrost was still thermally active in ...", "which was" eliminated

Figure 3: adjusted to " ... conditions (horizontal light-gray bar) (a), ... "

Line 171: as explained before, we prefer to stick to "hot-water drilling" which is more precise than "thermal drilling".

Line 190: changed into "devastating"

Line 201: done

Line 202: changed into "concern"

Line 204: done

Figure 4: data sources are added

Line 205: changed into "methodology"

Line 240: adjusted into " … field remained similar … "

Figure 5: north arrow and scales are added to every section

Line 253: "shown" changed into «used as background"

Figure 6: We decided to keep the colors as is.

Line 287: changed to "- 0.14 m/a" ("-" used consistently in the text)

Line 324: «clearly» eliminated

Line 342: adjusted into "The periglacial part of Gruben rock glacier with its ice-rich permafrost shows …"

Line 349: adjusted

Line 352: now always in two words

Line 367: this refers to the previous sentence with reference

Lines 370/71: done

Line 376: "very" deleted. The quantification of ice contents (by geophysical soundings) is mentioned in the same paragraph (line 380 ff).

Lines 376, 378: done

Line 379: added: "hydraulic and thermal" connectivity

Lines 383/84: done

Line 386: adjusted to «Intermittent flow acceleration could …".

Line 392: adjusted into " … - since the Little Ice Age a marked reorientation of the stress and flow fields." Following sentence adjusted

Lines 399/400: adjusted

Lines 407 – 411: simplified and rewritten as follows: "During its maximum LIA advance, the glacier with its cold margins must have exerted compressive stresses to the upper part of the rock glacier. This may have pushed ice-rich frozen debris upslope, towards the talus at the foot of the Outer Rothorn crest. Flow trajectories in this former contact zone are presently in a near-opposite direction, i.e. from the talus towards the now ice-free topographic depression (overdeepening) around lake 3.

Line 415: changed into "… as its debris-free part, but … "

Line 424: rewritten as "Even under similar thermal conditions (negative annual surface temperatures), the difference between the rock glacier and the debris-covered glacier is striking."

Line 433: done

Lines 441, 442, 443, 451, 452, 454, 455: adjusted according to the recommended version

Line 457: rewritten as " … so far provided protection against … "

Line 461: rewritten as " … developments, rock glaciers with their creeping frozen materials constitute persistent elements of cold mountain landscapes."

Line 477: adjusted

The study provides a) a review of existing measurements at the well investigated Gruben site in the Swiss Alps, b) extends the existing surface elevation change and velocity measurement to the recent time, c) presents some new in-situ measurements, and d) discusses based on the long term data the glacier-permafrost relation. This is one of the most comprehensive study on this topic presenting of the longest and most detailed time series on rock glaciers and is therefor of high interest and well suited for the journal. The study should ultimately be published, but there are several shortcomings which need to be addressed before publication.

Thanks for this constructive feedback.

*General comments*

Rock glacier vs. rockglacier: I understand the reason behind using rockglacier in one word and I am also fine with that. However, internationally writing "rock glacier" is much more common and I will use this spelling in my review.

We now write "rock glacier" in two words throughout the text.

Sections 1&2 (Intro and Study site)

The introduction section and Gruben site section should be better structured and in particular the section about the Gruben site could be more focussed. Some information about the study site is in the intro some in the specific section. However, some of the information of the Gruben site are needed to understand the rational of the study. Many information in the Gruben site section is quite old and probably already captured in Haeberli et al. (2001). My suggestion is to shorten the Gruben site section and focus on the relevant information for this specific study. It would also facilitate the understanding if the overview figure 1 and the geomorphological figure (Fig. 4) would be shown side by side. Maybe the permafrost zonation can be included in Fig. 4 (though you would need to change the colours) or shown in the supplement.

We decided not to shorten the text and leave the structure as is. By this, we provide all the facts from previous studies without having to search for the details in many other publications (like a review). It also makes it easier to understand figures 1 and 2, which we want to leave as they are (otherwise they will become too overloaded).

It is also not clear how you distinguish between the debris-covered glacier, the glacier affected rock glacier and the periglacial rock glaciers. A clarification is crucial are there are different options regarding the rock glacier origin as the authors are well aware of. Some relevant information is given in the results section but the information is needed earlier to be able to understand.

In the caption of Figure 1 we write that the blue line indicating the outline of Gruben glacier in the contact zone with the Gruben rock glacier at the time of the Little Ice Age (LIA) is approximate. We add in the text that the position of this somewhat diffuse contact zone is defined by (a) the clear margin of the debris-free glacier as indicated in the first reliable topographic map (1889; cf. supplementary material S4, S5 and S6), (b) the limits of exposed

massive ice as documented on the annually flown airphotos, and (c) the direction of the flow trajectories which lead from the talus at Outer Rothorn to the rock glacier front; see supplementary material S3).

The methods need to be more in depth described and quantitative uncertainty estimates provided (and not just referenced to other studies). Only with an uncertainty assessment you can state that the obtained results are within the "range of measurements uncertainty" (L. 241f) or are significant. The uncertainty/accuracy should be assessed by investigating the results over stable terrain, and for the recent time by comparison to existing in-situ measurements.

Agreed. The methods are well detailed in several publications that are mentioned, but we will add the quantified uncertainty in the methods chapter and link to Kääb et al. 1997 and Brunner 2020. "The error (root mean square) of single displacement measurements is about +/- 0.3 – 0.4 m or, in the case of a 5-year interval between two photo missions, about +/- 0.06-0.08 m/a and the error of the vertical changes is estimated to +/- 1 m or, +/- 0.17 – 0.2 m/a (Kääb et al. 1997; Brunner, 2020). If a large number of measurements are analyzed in combination (as e.g. in Figure 8), the statement is an order of magnitude more accurate (+/- 0.006 – 0.008 m/a)."

It needs also be clearly stated what was done in this study and which is based on earlier work. Please also clarify why you start in 1994 while the study by Kääb et al. (1997) provided data until 1995. How do the data from 1994 and 1995 match?

This is a valuable comment. Previous work is summarized in the introduction part and again consulted in the discussion. Our methods and results are detailed in the methodology and results chapters. The focus of this study first was more on the recent dynamics of the rock glacier and on the comparison to the debris-covered glacier tongue development. The latter was not investigated by Kääb et al. and therefore the selection of DTMs and orthophotos was made independently. With compiling the first draft of the paper, we realized that we have to include the previous data on the development of the rock glacier and the glacier and show the full picture.

We now added the data by Kääb et al. (1997) to Figure 5.

A quantitative uncertainty estimate is also beneficial for the results section where at least the most important numbers should be given along with the uncertainty range. The values given should also be more precise where possible (e.g. "are in about the same range of -0.1 to -0.5 m/a" – is there any difference at least in tendency or remained the surface lowering the same?, but this is only one example, there are several others).

The quantified uncertainty for horizontal and vertical changes is repeated in the results chapter (see also comment above).

The discussion would benefit from putting the results more into the context of the current rock glacier research worldwide. One example from Tien Shan is given but there are several other suitable examples from other parts of the world (though not from one of the authors).

A number of most recent quantitative studies were added, especially concerning long time series, quantitative glacier-permafrost investigation and absolute age dating.

Moreover, I ask the authors also to consider work by other researchers on the similar study site. I am aware of the different opinions about the origin of rock glaciers by different groups. In particular therefor it is important not to disregard but discuss relevant work by others (e.g. Whalley, W. B.: Gruben glacier and rock glacier, Wallis, Switzerland: glacier ice exposures and their interpretation, Geogr. Ann. A, 102, 141–161, doi:10.1080/04353676.2020.1765578, 2020.) but critically

We agree that the publication by Whalley (2020) must be mentioned, make reference to the corresponding visual observations of massive ice exposures in the former contact zone, and confirm again that there is no indication from measured facts of buried surface ice masses in the periglacial part of the rock glacier (cf. Haeberli. 2021 in a recent cryosphere discussion). We would like to leave it at that, because the publication by Whalley in Geografiska Annaler does not report any additional measurements and seems to ignore the state of knowledge at the site as summarized by Haeberli et al. (2001; this publication was a product of the Swiss National Research Programme 31). It is also evident that Whalley has not been aware of the ongoing monitoring activity within the framework of national and international climate-related observational programs (PERMOS, GTN-G of GCOS/GTOS).

Rather than on speculative, "either-or"-type "opinions" or "beliefs" concerning simplistic landform "origins", the focus of modern, quantitative and comprehensive glacier and permafrost research in cold mountains is on measured facts related to material properties, physical conditions and resulting processes/interactions as drivers of environmental and landscape evolution under conditions of global warming. This is now explicitly formulated already in the introduction.

Reference is now also made to the comprehensive geophysical investigations on the higher part of the Gruben glacier tongue by Kulessa (2009).

I have not counted in detail, but there are many self-citations. This is okay as the authors have done most of the work at the study cite but putting the own work better into the context of existing knowledge would also reduce the self-citation ratio.

Thanks and agreed. When extensive, long-term measurement and monitoring work at the Gruben site was initiated roughly half a century ago, this effort constituted one of the earliest comprehensive and quantitative investigations on glacier-permafrost-lake relations and interactions in cold mountains under conditions of rapid global warming. In this sense, there is a historical dimension to this long-term, interdisciplinary and application-oriented cooperation on behalf of political authorities and scientific organizations. We consider it adequate to document how our fact-based knowledge and understanding evolved through time. The interest in quantitative investigation of complex glacier-permafrost contacts and relations using modern technologies indeed sharply grew during recent years. In order to

reflect this, we added a number of modern quantitative studies, especially concerning long time series of permafrost creep and glacier-permafrost relations.

*Detailed comments*

Title: Write "5" in letters "five"

done

L 54: There have been many more recent studies related to rock glacier creep. Please cite one or two mere recent ones in addition to Roer (2007)

Mention is made of an additional number of most recent studies

L 55: Be more specific: Are the typical depths of the shear horizon of around valid for the rock glaciers on Earth or for the Swiss Alps were measurements are available?

Shear horizons can only be determined through precise borehole deformation experiments. Such information is available from various places in the European Alps.

L 88: "periglacial and glacier-affected parts": This should be explained in the introduction to be able to understand the purpose.

Agreed - we now write: " … compare the creep characteristics of the perennially frozen rock glacier ("periglacial part") and its former contact zone with  the polythermal glacier ("glacier-affected part") …

L 94f: I suggest to first summarise the most important characteristics of the Gruben site and then refer to Haeberli et al. (2001) for more details (see also my general comment above).

Agreed. We deleted the first sentence.

L 104f: Provide a reference for the statement of the temperatures during LIA and the precipitation.

Reference added.

Figure 1: This is a key figure to understand the situation. It is in general good, but could and should be further improved. I suggest including a legend with the most important symbols/letters (e.g. the lines) of the figure (or write the letter and numbers in a table associated to the figure. This would make the figure more easily understandable. Moreover, I suggest to add some symbols (e.g. to indicate the rock glacier fronts etc.). I can imagine that it is for a non expert no easy to identify to which form the letters are referring to exactly (or show the figure side by side with Fig. 4, see above). Moreover, how was the approx. LIA extent and the flow during LIA determined?

The LIA extent and the flowline reconstruction is based on the first reliable topographic map from 1889 which is added in the supplementary material 4 (see also the formulations in the figure caption and in the text).

Figure 2: Please include the letters, numbers and lines in the legend. Indicate the elevation of the mountain peaks. Also include the rock glacier and glacier outlines as this is very important information. See also my general comment above.

Agreed. We will add the elevations of the peaks and the outlines. If there is enough space, we will also add letters and numbers to the legend.

L 124: What are the modelling results of the two global models for the site? Please include the sites with BTS measurements and clearly mention how well they fit to the models.

In the supplementary material S7 and S8, we now provide two more high-resolution simulations for comparison. We briefly comment on the general agreement with low-resolution global/hemispheric models. The position of BTS measurements, geophysical soundings and shallow core drilling is documented in graphs from earlier publications as reproduced in the supplementary material S1, S2 and S3. The dashed white lines in Figure 2 indicating the transition to permafrost-free terrain are drawn using the information from these measurements; they illustrate the degree of local agreement/disagreement with the permafrost model by Böckli et al. as applied to the Gruben site.

L 148: Are these permafrost landforms in Fig. 1? If yes, please refer to it more clearly. If not: Include in Fig. 1.

Yes. This is now made clear in the caption of Figure 1 as well as in the text.

L 155: Who determined the 0.15 times surface velocity at the front?

This comes from Kääb et al. 1997; Reference is added.

L 169ff: The reference is now 45 years old. Others are also now the most recent ones. Please state critically if the statements are still true considering the temperature increase the last decades.

Englacial temperature measurements have not been repeated and much of the largely debris-free glacier tongue has melted away during the past 50 years. Temperatures within the remaining debris-covered glacier tongue are estimated to be slightly colder than in the adjacent rock glacier permafrost, because of more shadow from the Inner Rothorn. This is now explained in the text.

L 210ff (section 3 data and methods): it is not clear from the section what was actually done within this study and which data was already processed by earlier ones (e.g. Kääb, 2005; Roer et al., 2005a; Brunner, 2020). Please clarify. Moreover provide the most important information about the processing steps also in this study. Most important: provide quantitative estimation of the accuracy. See also general comment above.

Done: see responses above.

Figure 5: I recommend the most common colour coding convention and present surface lowering in red and elevation gain in blue.

Agreed and done.

I ask to provide a more detailed subdivision of the elevation changes, so that more details are visible (e.g. >1.0, 1.0-0.75, 0.75 – 0.50, 0.50-0.25, 0.25 – 0.10, 0.10 - - 0.10, -0.10 - -0.25 …). I do not see any white colour. I suggest showing and not show this range transparent. Please add the info about where are the different parts of the glacier and rock glacier. I ask you also to show the elevation change outside the glacier and rock glacier area which enables to visually assess the accuracy (either in this figure or in the supplement). It would certainly also of high interest to know how the parts of the south-eastern moraine (where the road leads to) changed over time (e.g. if there is an ice core I would expect at least some surface lowering).

We decided to keep the subdivision of the elevation changes as is, as well as the colour coding. The range between -0.15 and 0.15 is transparent, as suggested. In addition we decided to show the data only for the selected landforms of interest and not for the surrounding areas, as it would make it more difficult to see the results, as several other processes occur around the landforms and on steeper slopes the quality of the data is reduced. More details kann be taken from Kääb et al. (1997) and Brunner (2000).

The road on the steep south-eastern slope of the LIA moraine constantly deformed and had to be repeatedly re-opened for construction work. The topographic changes at this specific site are therefore heavily influenced by human activity. Exposures of massive ice have not been observed here.

This paper investigates about five decades of kinematic monitoring at the Gruben rock glacier and glacier site in the Valais, Swiss Alps. The study tries to better understand the evolution between a more permafrost influenced structure and the polythermal glacier part, which form several complex geomorphological forms with different vertical and horizontal changes. This study is an excellent example what efforts of long-term observations can provide to better understand geomorphological landforms and their process-based behaviors. The study is very well prepared and written and the history of the study is carefully compiled.

Thanks.

General comments:

Figure 1 and 2 could be joined into one single figure and the color scale of the permafrost distribution model of Böckli et al. 2012 could be strongly reduced to a very light transparency level. If the authors do not want to change this, then the figure 2 should be deleted and the dashed white lines should be integrated in figure 1.

We decided to keep both Figures separate with some adaptations (see above), as they allow to understand the geomorphological setting (Figure 1) and the (modelled) permafrost distribution (Figure 2). A combination would be too overloaded and the permafrost distribution is too important to be relegated to the supplementary material.

It is understandable that the authors have not included the old measurements of Kääb et al. 1997. However, this study is somehow missing this important information and the study would strongly profit, if the old information of Kääb et al. 1997 could be included in Figure 5 to particularly show the whole investigated period. I do not think that this is a duplication of information, but readers would probably like to have access to full five decades and not only the new data since 1994 to 2016.

Agreed. The data 1970-1995 by Kääb et al. (1997) are now added to Figure 5 (5A).

Specific comments:

Line 124: additional citation: there are more permafrost models with higher resolution such as Kenner et al. 2019 (Kenner, R.; Noetzli, J.; Hoelzle, M.; Raetzo, H.; Phillips, M., 2019: Distinguishing ice-rich and ice-poor permafrost to map ground temperatures and ground ice occurrence in the Swiss Alps. Cryosphere, 13, 7: 1925-1941. doi: 10.5194/tc-13-1925-2019

Thanks. Kenner et al. (2019) is now cited and examples of other simulations for the Gruben site are provided in the supplementary material S7 and S8.

Line 135: please add some references how sediment rates are determined, if the authors want to estimate the sedimentation during the whole Holocene and the development of the rock glacier and glacier evolutions. How would certain sediment rates fit with their own estimates to create the current periglacial and glacial environments?

No sedimentation rates were determined. Quantitative estimates would be extremely vague

under conditions of repeated lake outbursts causing efficient episodic sediment evacuation through large debris flows. Qualitatively, the total mass of sediments in the cirque can be said to be quite enormous as solid bedrock underneath the glacier bed is only at a depth of > 100 m as documented by borehole resistivity soundings (Haeberli and Fisch 1984).

Line 169: at many places numbers are written like 10m instead of 10 m -> please correct all these numbers in the whole paper

Thanks, checked and adjusted.

Figure 5: please change in all diagrams blue versus red. Red is more suited to negative values and blue more to positive ones

Agreed and done.

Figure 6: how was the blue line in the figure distinguishing rock glacier from glacier affected part in the figure and why is there no connection between topographical features in the map and the blue curve. Please give some more details

We now emphasize more clearly that the limits of the contact zone are diffuse and that this may most probably be due to the fact that the cold glacier margins did not efficiently affect the surface topography of the rock-glacier permafrost (see also comments on earlier topographic maps in the supplementary material S4 and S5).

Line 331: giving a retreat of the no debris covered glacier part is somewhat problematic as this part is not really the glacier tongue as it is still connected to the debris covered part of the glacier and showing a retreat of this part is not very convincing. In addition, giving the full retreat of 370 m is ok, but showing the annual mean values of 17 m does not make sense, as one knows that glacier retreat can be highly variable and if the individual annual values are not measured, the annual values should not be provided.

This is correct and now also formulated in this sense. With the changes of the glacier geometry, flowline orientations also changed, turning towards lake 3. As a consequence, what was originally the orographic right margin more and more developed into some sort of a broad terminal margin. The given numbers are averages over the observed time interval and help defining response magnitudes in comparison with the decaying debris-covered glacier part, the now vanished dead ice in the contact zone with the rock-glacier permafrost and with the periglacial perennially frozen debris.
* * *
Entire document:

Check for space between numbers and units: done

a or y for "per year" consistent throughout text: done -  now always m/a

hyphening consistent: checked
* * *
In addition to the reviewer comments we will include a few sentences on available data from a permanent GNSS station on Gruben rock glacier:

Line 229 added: Further, permanent GNSS data are available for the Gruben rock glacier from 2012 onwards. The fixed station provides high-resolution data (seasonal to sub-seasonal) on surface deformation (Beutel et al. 2021).

Discussion: Consistent movement rates are derived from photogrammetry and GNSS data, which fit well with the data of the permanent GNSS, situated between the bedrock riegel and the rock glacier front (Beutel et al. 2021).
…
Continued thermal and kinematic monitoring at this well-documented site and the detailed analysis of the high-resolution data from the permanent GNSS (Beutel et al. 2021), can shed more light on the short-term and longterm changes, as well as on involved processes.

---

## Author Response (AR2)

**Response to reviews**

Glacier-permafrost relations in a high-mountain environment: Five decades of kinematic monitoring at the Gruben site, Swiss Alps

*Isabelle Gärtner-Roer, Nina Brunner, Reynald Delaloye, Wilfried Haeberli, Andreas Kääb, and Patrick Thee*

We thank the reviewer and the editor for the constructive comments. We revised the manuscript again based on the feedback. Please find below the individual feedback to all comments.

Best,

Isabelle Gärtner-Roer, with all co-authors

From the editor:

Dear authors
Many thanks for your revisions. You have adequately addressed most review comments – apart from some very valuable comments by Reviewer #2. In particular, I do not find the accuracies/uncertainties in the paper – though you state you would have added those, e.g. in the Methods sections. As far as I can see, only for the GNSS data the measurement errors are given.

This is correct and relates to a mistake on our side. We added the text in the last "response to reviews", but forgot to add it to the manuscript. The following text is now added in chapter 3 (Geomatics):

"The error (root mean square) of single displacement measurements is about +/- 0.3 – 0.4 m or, in the case of a 5-year interval between two photo missions, about +/- 0.06-0.08 m/a and the error of the vertical changes is estimated to +/- 1 m or, +/- 0.17 – 0.2 m/a (Kääb et al. 1997; Brunner, 2020). If a large number of measurements are analyzed in combination (as e.g. in Figure 8), the statement is an order of magnitude more accurate (+/- 0.006 – 0.008 m/a)."

Also, as pointed out by Reviewer #2, using the qualifier «significantly » requires providing the underlaying statistics. → adapted in the text
Overall, I fully agree that your article comprehensively describes the monitoring of this extraordinary site, which allowed to understand the complex kinematics and glacier-rock glacier interactions. As pointed out by Reviewer #2, and being not familiar with the very many, different studies, I am also left wondering what was done in this study and which is based on earlier work.

We carefully reconsidered this aspect again and came to the conclusion that already the first lines (10-15) of the abstract clearly define what the new information is. The interpretation of these new measurements must relate to the scientific background from the comprehensive field investigations carried out during roughly half a century. This is carefully documented in the lines 69-83 of the introduction. The purpose of our paper - points (i) to (iii) - is then defined on the lines 90-99 of the Introduction. In the results chapter we show all the new data, beside the photogrammetric analyses

from Kääb (1997) for the long-term development (Figure 5), which was the explicit wish of the reviewers.

Nevertheless, I will accept the manuscript once you have addressed the above (and below) comments.

Thank you very much.

Some minor points
Please consider the comments by Reviewer #1 on the revised manuscript. → done (see below)
Please include an explanation into the captions of Figs. 4 and 5 what the numbers on the axes mean, something along the lines: "Axes are labeled with Swiss coordinates (in meters; CH 1903)." → explanation added
I recommend using elevation rather than altitude (except for things that fly). → done
Line 223: Please use past tense when describing the state of lakes in 2016. → done

Jürg Schweizer.

Review by Lukas Arenson:

Manuscript should be accepted ad is.

Thanks for revising the manuscript. The comments made by the reviewer on the initial version of the paper have been addressed satisfactory in my view.

Thank you very much.

Some very minor editorial corrections:
- Line 56: delete in at the beginning of the line → deleted
- Line 98: add comma after cold → done
- Line 101: "perennially frozen" is not needed because an active rock glacier is by definition in a permafrost environment and therefore must contain ground ice. And "perennially" is probably no longer accurate anyway. → deleted
- Line 133: FOEN is missing in the references → reference added
- Line 146, and other places: there are still many places where the space between the number and the unit is missing. → checked & done
- Line 420: long-term → done

From the TC MS records:

With the next revision, please add the copyright icon to the figure source of Figure 4.

    → done